# Development of aerosol activation in the double-moment Unified Model and evaluation with CLARIFY measurements

Hamish Gordon[1,2], Paul R. Field[1,3], Steven J. Abel[3], Paul Barrett[3], Keith Bower[4], Ian Crawford[4], Zhiqiang Cui[1], Daniel P. Grosvenor[1], Adrian A. Hill[3], Jonathan Taylor[4], Jonathan Wilkinson[3], Huihui Wu[4], and Ken S. Carslaw[1]

[1]School of Earth and Environment, University of Leeds, LS2 9JT, United Kingdom
[2]Engineering Research Accelerator, Carnegie Mellon University, Forbes Avenue, Pittsburgh 15213, United States
[3]Met Office, Fitzroy Road, Exeter, EX1 3PB, United Kingdom
[4]Department of Earth and Environmental Sciences, University of Manchester, M13 9PL, United Kingdom

*Correspondence to:* Hamish Gordon hamish.gordon@cern.ch

**Abstract.** Representing the number and mass of cloud and aerosol particles independently in a climate, weather prediction or air quality model is important in order to simulate aerosol direct and indirect effects on radiation balance. Here we introduce the first configuration of the UK Met Office Unified Model in which both cloud and aerosol particles have 'double-moment' representations with prognostic number and mass. The GLOMAP aerosol microphysics scheme, already used in the HadGEM3 climate configuration, is coupled to the CASIM cloud microphysics scheme. We demonstrate the performance of the new configuration in high-resolution simulations of a case study defined from the CLARIFY aircraft campaign in 2017 near Ascension Island in the tropical south Atlantic. We improve the physical basis of the activation scheme by representing the effect of existing cloud droplets on the activation of new aerosol, and we also discuss the effect of unresolved vertical velocities. We show that neglect of these two competing effects in previous studies led to compensating errors but realistic droplet concentrations. While these changes lead only to a modest improvement in model performance, they reinforce our confidence in the ability of the model microphysics code to simulate the aerosol-cloud microphysical interactions it was designed to represent. Capturing these interactions accurately is critical to simulating aerosol effects on climate.

## 1 Introduction

Shallow marine clouds are an important source of uncertainties in climate forcing and sensitivity. Representing aerosol effects on these clouds is a priority for climate modelling efforts worldwide. In this paper, we describe model simulations in a $2°$ by $2°$ region of the tropical south Atlantic Ocean near Ascension Island. The simulations are performed with the UK Met Office Unified Model (UM), with double-moment aerosol microphysics driving double-moment bulk cloud microphysics at $500\,\mathrm{m}$ horizontal resolution. The cloud and aerosol microphysics parameterizations form, or are intended to form in future, part of the atmosphere model code used for climate simulations in Coupled Model Intercomparison Project (CMIP) experiments and for operational weather forecasts across the Unified Model partnership. The rest of the model code is also used in the HadGEM3-GC3.1 configuration for CMIP6 and in current operational numerical weather prediction configurations. As well as testing the

aerosol and chemistry component of the model at a higher resolution than has been attempted before, we study and suggest improvements to the performance of the aerosol activation scheme. We evaluate the second day of a two-day long simulation against CLARIFY (CLouds and Aerosol Radiative Impacts and Forcing) aircraft measurements on 19 August 2017.

A series of recent field campaigns (Zuidema et al., 2016) have focused on the tropical south-east Atlantic ocean, which hosts one of the planet's largest stratocumulus decks, and is the destination for much of the biomass burning aerosol that originates from central and southern Africa. The prevailing winds, which are south-easterly in the boundary layer and easterly in the free troposphere, advect smoke over the ocean where slow subsidence causes the smoke to mix with the clouds. The smoke can have large direct, semi-direct and indirect radiative effects on the regional climate (Costantino and Bréon, 2013; Lu et al., 2018; Gordon et al., 2018). Here we focus on the indirect effect of the aerosols.

In a previous study, Gordon et al. (2018) evaluated aerosol transport and microphysics in global and convection-permitting simulations of the south-east Atlantic. They built on earlier work with the same aerosol microphysics scheme employed at high spatial resolution by Planche et al. (2017). In this paper, we start from an updated version of the same atmospheric model, and increase the resolution further, to approach the cloud-resolving scale. We also increase the sophistication of the cloud microphysics scheme from single-moment to double-moment, in order to study aerosol-cloud interactions in more detail. The resulting model is the first configuration of the Unified Model with fully double-moment aerosol and cloud microphysics. Double-moment microphysics, including prognostic cloud droplet number concentration, is important to enable good representations of processes such as aerosol activation and droplet settling at high spatial and temporal resolution. The coupling to the double-moment interactive aerosol microphysics scheme enables aerosol-induced variability in the droplet number concentration. We evaluate the aerosol and cloud microphysics in the new model configuration in this paper, paying particular attention to shortcomings of the simulations that are specific to aerosol-cloud interactions or to simulating aerosols at high spatial resolution. We also highlight some underlying issues with aerosols in the coarse-resolution global climate model that drives our regional simulations, which we will address in future work.

Compared to single-moment cloud microphysics schemes, double-moment schemes have been shown previously to improve the representation of stratiform rain in NWP simulations (Morrison et al., 2009) and to reduce a range of biases in high-resolution climate models (Seiki et al., 2015). The CASIM (Cloud AeroSol Interacting Microphysics) double-moment microphysics scheme we use here is that published previously (Shipway and Hill, 2012; Grosvenor et al., 2017; Miltenberger et al., 2018; Stevens et al., 2018) and Furtado et al. (2018). In the last of these manuscripts, Furtado et al. (2018) evaluate CASIM for deep convective clouds and compare it to a reduced, single-moment version of the same scheme, and to the different single-moment cloud microphysics scheme (Wilson and Ballard, 1999) used in the operational version of the model. For the case they study, the CASIM double-moment microphysics initially performed better than the single-moment schemes, although these gave comparable performance to the double-moment scheme when tuned.

In climate simulations with resolutions coarser than around 0.5 degrees, updraft velocities in shallow clouds are almost entirely unresolved and convection is parameterized. The activation of aerosols to form cloud droplets requires the supersaturation of water vapor relative to aerosols and hydrometeors to be diagnosed or parameterized. Typically, supersaturation is calculated by imposing an updraft speed (or a distribution of updraft speeds) derived from diagnostics of the sub-grid turbulence rather

than the grid-box mean updraft speed, which is close to zero. A single cloud droplet number concentration per grid box is thus produced and used in the prediction of rain rates (via an autoconversion parameterization) and cloud albedo.

As the resolution of the simulation is increased into the 'terra incognita' or 'gray zone' (Wyngaard, 2004), a higher fraction of the turbulence in the boundary layer is resolved, until, at the large eddy simulation (LES) scale of order $10\,\mathrm{m}$, we assume for the purposes of this paper that the spatial variability of prognosed updrafts would be a good representation of reality. The turbulence starts to be resolved when the effective grid resolution is below about four times the height of the boundary layer (Honnert et al., 2011), typically $4-8\,\mathrm{km}$. There is therefore a point at which it is no longer necessary to use an updraft speed diagnosed from sub-grid turbulence in the activation scheme, and the grid-box mean can be used instead. Prior CASIM simulations (Grosvenor et al., 2017; Miltenberger et al., 2018; Furtado et al., 2018) have also used grid-box mean updraft speeds to activate aerosols at horizontal resolutions ranging from $250\,\mathrm{m}$ to $1\,\mathrm{km}$, and a similar approach has been taken in Regional Atmospheric Modeling System (RAMS) and some Weather Research Forecasting (WRF) simulations (Saleeby and Cotton, 2004; Thompson, 2016). The scale invariance of activation schemes has been tested before down to horizontal resolutions of $1\,\mathrm{km}$ (Possner et al., 2016). However, these resolutions are still much coarser than typical LES resolution, and therefore the full variability in updraft speeds will not be resolved. By comparing near-cloud-resolving and LES simulations, Malavelle et al. (2014) developed a bootstrapping parameterization which enables the estimation of the fraction of the variance in updraft that is resolved.

In existing clouds, activation of new droplets will often be negligible, but not always, for example if the updraft strengthens towards the top of the cloud. In a detailed model, Pinsky and Khain (2002) suggested that in-cloud activation leads to a bimodal cloud droplet size spectrum, and is an important factor in accelerating rain formation by broadening the droplet spectrum (Segal et al., 2003; Heymsfield et al., 2009). In a deep cloud case study, the large eddy simulations of Fridlind et al. (2004) suggested that aerosol concentrations in the boundary layer were too low to explain observed droplet number concentrations, and therefore aerosols must be entrained from the mid- and upper troposphere and activated inside the clouds. More recent detailed modeling studies have continued to investigate secondary activation (Khain et al., 2012; Fan et al., 2018). To capture fully effects of in-cloud activation on the droplet size distribution, a detailed microphysics scheme is needed, such as a size-bin-resolving scheme or a super-droplet model, and to accurately simulate the full dynamical response of deep clouds to aerosols, very high spatial resolution may be needed, which in turn requires relatively short timesteps. If the model timestep is shorter than the 'relaxation timescale', the timescale for supersaturation production to be balanced by condensation onto existing droplets, a bias will occur unless supersaturation is represented as a prognostic variable (Khain and Lynn, 2009; Fan et al., 2012; Lebo et al., 2012; Grabowski and Jarecka, 2015; Grabowski and Morrison, 2017). This timescale is typically a few seconds, depending on the vertical velocity and droplet spectrum. Prognostic supersaturation is desirable in simulations requiring accurate calculations of the latent heat released by condensation (Grabowski, 2007), but the short timesteps needed mean it is very expensive to treat supersaturation prognostically (Arnason and Brown, 1971; Morrison and Grabowski, 2008), although this can be mitigated to some extent with semi-analytic approaches (Clark, 1973; Hall, 1980). The relaxation timescale is short (less than 10 seconds) in the case of thick, polluted clouds with many or large droplets and/or low updraft speeds, and long in the case of very clean clouds or strong updrafts (Kogan and Martin, 1994). If, however, the model timestep is longer than the relaxation timescale

(and we verify that this is the case in our study), one can then assume that supersaturation produced during a timestep leads to condensation immediately, so that the relative humidity is 100% at the end of the timestep. This assumption, termed 'saturation adjustment', may lead to biases in the latent heat released by condensation and in the evaporation of clouds. However, it is much simpler and computationally cheaper than treating supersaturation prognostically, and it works well when updraft velocities are not fully resolved, as in weather prediction and climate models.

In our CASIM microphysics scheme, saturation adjustment is applied. It has sometimes been assumed that prognostic supersaturation is generally part of double-moment microphysics schemes (e.g. Guichard and Couvreux, 2017). However, Shipway and Hill (2012) compared several single and double-moment bulk microphysics schemes including CASIM with a bin microphysics scheme in a single-column framework. The bin scheme treated supersaturation prognostically while most of the bulk schemes did not. The double-moment bulk schemes with saturation adjustment they tested were in closer agreement with the bin microphysics scheme than the single-moment schemes. Their conclusions were substantiated further by Hill et al. (2015). While useful, prognostic supersaturation is not essential for a double-moment microphysics scheme to improve on a single-moment scheme.

In our model, on each timestep the activation scheme is re-run assuming there are no existing droplets. If the new droplet concentration is greater than the existing droplet concentration, the old droplet concentration is overwritten by the new. Changes to the cloud fraction in the gridbox may also change the droplet concentration, as discussed later. A similar procedure is followed in the widely used Morrison and Gettelman (2008) scheme. In this study, we follow the suggestion by Korolev (1995); Ghan et al. (2011), and others to improve on this procedure by accounting for existing cloud droplets when new droplets are activated, assuming a supersaturation that results from a balance between production (updraft) and loss (condensation on existing droplets). This assumption can be seen as a natural extension of the saturation adjustment assumption, and should apply in the same conditions. When it was tested in WRF-chem with Morrison et al. (2009) cloud microphysics, Yang et al. (2015) found that it improved simulated wet scavenging. Our implementation is aware of and consistent with our simulated sub-grid cloud fraction. We emphasise that we seek here to improve the accuracy of our existing model, but our improvements will not enable it to compete with detailed studies of aerosol-cloud interactions that employ spectral bin microphysics or prognostic supersaturation.

We attempt to improve the activation scheme further by considering unresolved sub-grid-scale updraft velocities. We examine the suitability of the parameterization of Malavelle et al. (2014) for our model, but we are not yet able to implement it explicitly, and instead we derive an ad-hoc correction factor suitable for our case study. We show, unsurprisingly, that accounting for existing cloud droplets in the activation scheme reduces the cloud droplet number while our correction factor for sub-grid-scale updrafts increases it. This may explain why previous studies with CASIM (Grosvenor et al., 2017) have successfully produced realistic droplet concentrations with realistic CCN, despite underestimating the updraft and ignoring existing droplets. We examine the implications of our improvements for the cloud droplet spatial distribution and size distribution and for rain formation. We evaluate our simulations against CLARIFY aircraft measurements in 2017 to confirm they are realistic and to identify directions for future developments.

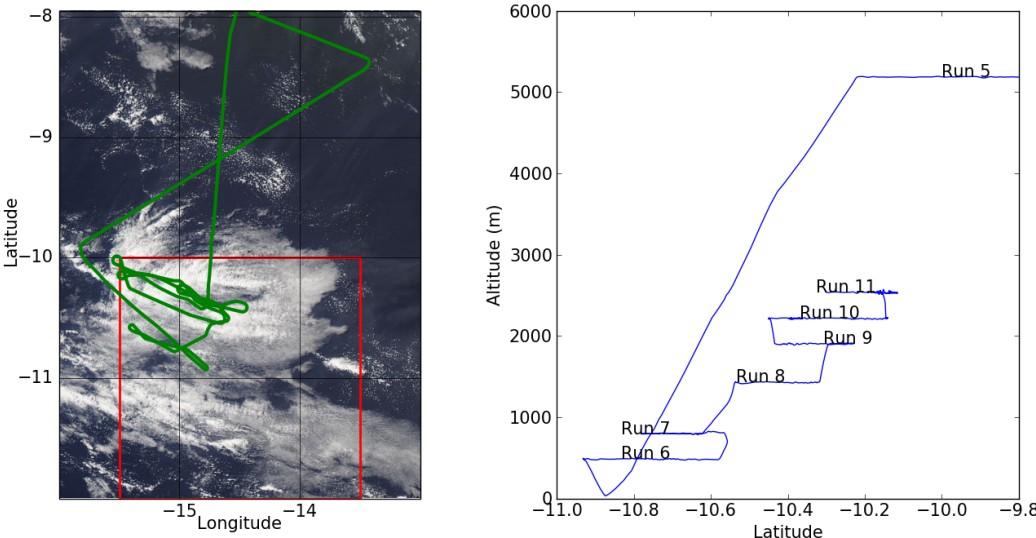

**Figure 1.** Flight pattern of the CLARIFY flight on 19 August 2017 used in the evaluations presented here, represented by the green line. The left plot is superposed on imagery from MODIS on the TERRA satellite that corresponds to the morning of the flight, showing the cloud deck that was sampled. The red box corresponds to the domain of the 500 m-resolution simulation. The right panel shows the path of the aircraft as a function of height and latitude.

## 2   Case study and aircraft measurements

The CLARIFY aircraft campaign took place from 16 August to 7 September 2017. The BAE-146 aircraft of the Facility for Airborne Atmospheric Measurements (FAAM) was based at Ascension Island during this period. Extensive sampling of biomass burning aerosol interacting with clouds was achieved during 24 flights usually of around 3.5-4 hours' duration. The aircraft flew in all directions around Ascension Island, according to where aerosol and cloud transitions could be identified or in order to pass under satellite tracks.

The research aircraft was fitted with a comprehensive suite of thermodynamic, radiometric, cloud physics and aerosol instrumentation for the CLARIFY field campaign. Ambient air temperature was measured using a non-deiced Rosemount/Goodrich type-102 total air temperature sensor. Atmospheric water vapour was measured using a WVSS-II near-infrared tunable diode laser absorption spectrometer fed from a standard flush mounted inlet (Vance et al., 2015). The temperature and humidity measurements are used to calculate relative humidity in cloud-free air. The temperature measurement in cloud was subject to significant wetting effects that led to a cold bias, as illustrated by Heymsfield et al. (1979). The derived in-cloud relative humidity is therefore set to 100% in this work. Zonal, meridional and vertical wind components were derived from the five-port turbulence probe located on the aircraft radome (Petersen and Renfrew, 2009; Barrett et al., 2019). On the transit at 5180 m altitude from Ascension Island to the cloud, the mean vertical velocity was $-0.010\,\mathrm{ms^{-1}}$. The small magnitude of this mean

velocity compared to the in-cloud vertical velocities we discuss later suggests the probe is sufficiently well calibrated for our analysis so we do not subtract any baseline offset from the observed updrafts in the evaluation we present in Section 6.

The size distribution of aerosol particles was measured using a Passive Cavity Aerosol Spectrometer Probe (PCASP, Droplet Measurement Technologies) for nominal diameters between about $0.1$ and $3\,\mu m$. We applied a complex refractive index of $r_i = 1.54 - 0.027i$, appropriate for biomass burning aerosol during the CLARIFY time period (Peers et al., 2019) and recomputed the bin boundaries for the PCASP instrument. This resulted in changes to the locations of bin centres, compared to the nominal values from the manufacturer, of usually around 5% but sometimes up to 20% in the diameter range below 1 μm. For smaller particles with diameters between 0.03 and 0.3 microns, we also used a Scanning Mobility Particle Sizer (SMPS). The total number concentration of aerosols with diameter larger than about $2.5\,nm$ were measured with a TSI 3786 Condensation Particle Counter (CPC). Aerosol data were only used when in cloud- and precipitation-free air. We determined this using the standard deviation of raw power on the Nevzorov total water content probe (Korolev et al., 2013), where a power greater than $3.0\,mW$ ($\sim 1.5 \times 10^{-4}\,gm^{-3}$) indicates cloud conditions, following Barrett et al. (2019). An additional safety window of $5\,s$ ($\sim 500\,m$) either side of positively identified cloud was applied to account for diffuse cloud edges and imperfect temporal and spatial synchronisation between probes and data recording systems. The cloud droplet size distribution (between 2 and $52\,\mu m$ diameter) was measured with a Cloud Droplet Probe (CDP) that was calibrated with a ten point bead calibration (Rosenberg et al., 2012). Precipitation-sized particles were measured using a 2D-Stereo (2DS) probe (10 to $1280\,\mu m$ diameter) and a Cloud Imaging Probe (CIP-100) probe ($100\,\mu m$ to $6.4\,mm$ diameter). Data from the CDP, 2DS and CIP-100 are combined to produce a composite PSD at a $1\,Hz$ sampling frequency following the method of Abel and Boutle (2012). Elsewhere in this paper, the cloud drop number concentration (CDNC) and liquid water content (LWC) are calculated using the CDP data only. For cloud measurements, in-cloud conditions were determined using a liquid water content threshold of $0.01\,g\,kg^{-1}$ calculated by integrating the cloud drop number size distribution from the CDP.

The most comprehensive sampling of a cloud feature took place on 19 August. The aircraft flew south of the island to sample a large precipitating cloud structure around $1.5°$ in size. Biomass burning aerosol was detected both within and just above the boundary layer. Cloud top height peaked at around $2.5\,km$ altitude. The cloud deck was sampled as shown in Figure 1, in a series of five straight-and-level aircraft trajectories along a line of strong radar echoes seen on the aircraft weather radar.

Taken together, the aircraft observations described later in the paper suggest that the boundary layer is decoupled or cumulus-like, with a stratocumulus cap above large precipitating cumulus clouds beneath. The observed stratocumulus clouds could also be detrained remnants of cumulus. In the simulations, the cumulus sometimes seems to extend up to the top of the boundary layer. However, the size of the cloud feature is larger than average for stratocumulus-to-cumulus transition clouds, and the deepest cumulus appeared to be organized linearly. Based on this and on geostationary satellite imagery (not shown) we would describe it as the beginnings of a fish rather than sugar, flowers or gravel (Stevens et al., 2019).

In order to validate our simulations, we are also able to draw on surface measurements from the ARM site at Ascension Island, and satellite observations of cloud droplet concentration and liquid water path. To obtain these, the same procedure as used by Gordon et al. (2018) is followed.

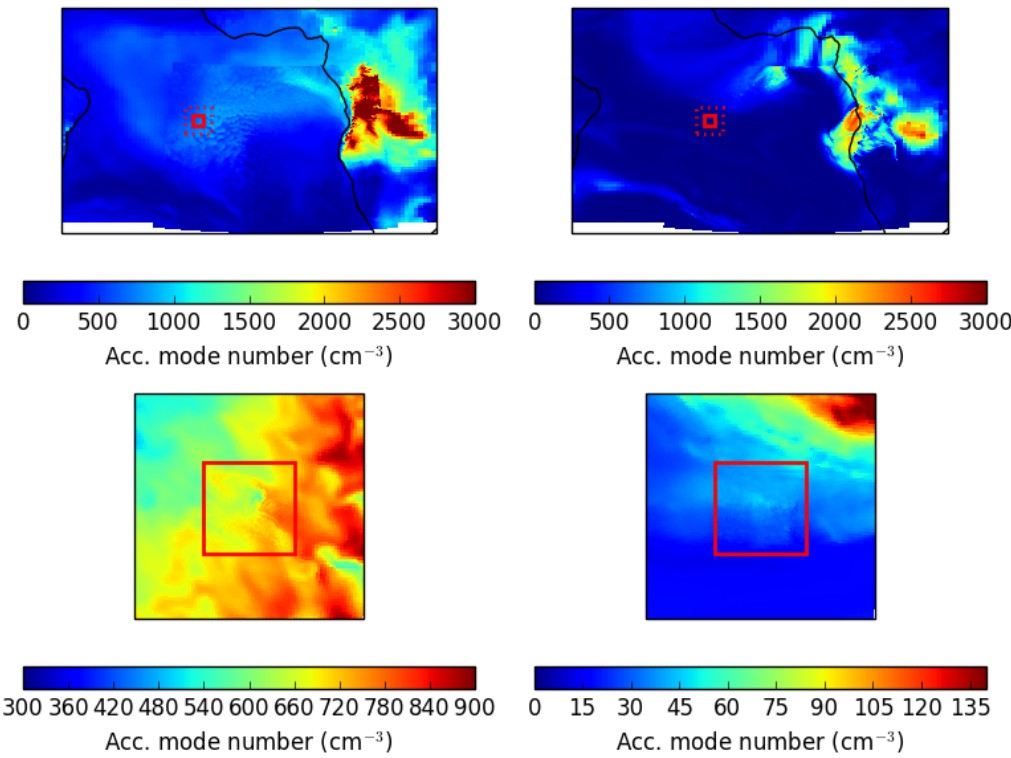

**Figure 2.** Simulated number concentration of accumulation-mode aerosol, (abbreviated as Acc-mode in the legend labels) in the boundary layer (averaged from the surface to 750m altitude) on the left and in the lower free troposphere (averaged from 2900 m to 3400 m altitude) on the right), in $cm^{-3}$, at midday UTC on 19 August. The 500 m regional domain is shown as a red square on all four plots; the top plots show the global and 7 km models with the domain of the bottom plots as a red dotted square, while the bottom show the 7 km model and the 500 m model.

## 3 Model setup

To establish a regional simulation at 500 m spatial resolution, we set up three configurations of the Unified Model (UM): a global model, a 7 km-resolution regional model, and the 500 m-resolution model. The code for all three models is almost identical, except for the cloud microphysics, as described later. The global model is used to produce lateral boundary conditions for the 7 km-resolution model, and this in turn provides boundary conditions to the 500 m model. The higher-resolution models do not feed back to the lower-resolution models (our setup is usually described as 'one-way nesting'). To illustrate the setup, simulated accumulation-mode aerosol number concentrations are shown in the 7 km model with the global model (top) and 500 m model (bottom) in Figure 2. The configuration of the three models is summarised in Table 1 and described in this Section.

**Table 1.** Summary of differences between model configurations used. Unless explicitly mentioned, affected by recent minor updates, or dictated by the name of the model configuration as described in the appropriate documentation paper, the code for global and regional simulations is identical. In the 'Cloud microphysics' row 1M signifies single-moment and 2M double-moment. In the 'Aerosol activation' row, ARG stands for the parameterization of Abdul-Razzak and Ghan (2000). This parameterization has two implementations, labelled DIAG when cloud droplet number concentration is diagnostic, and PROG when it is prognostic. The microphysics sub-step is marked with an asterisk because we wish to emphasise that the sub-step is not applied to condensation or evaporation, which are treated by the cloud parameterization and not by the microphysics scheme. However, it does apply to precipitation-related processes such as autoconversion and accretion, for example.

| Configuration or parameterization | Global | 7 km model | 500 m model |
| --- | --- | --- | --- |
| Configuration name | GA7.1 | RA1 | RA1 |
| Number of lat,lon grid cells | 324x432 | 450x670 | 450x450 |
| Timestep | 20 minutes | 2 minutes | 20 seconds |
| Microphysics sub-step* | 2 minutes | none | 10 seconds |
| Number of vertical levels | 85 (to 85km) | 140 (to 40km) | 140 (to 40km) |
| Code version | 11.2 | 10.8 | 11.3 |
| Cloud microphysics | 1M (Wilson and Ballard, 1999) | 1M (Wilson and Ballard, 1999) | 2M (CASIM; Shipway and Hill (2012)) |
| Aerosol microphysics | GLOMAP (Mann et al., 2010) | GLOMAP (Mann et al., 2010) | GLOMAP (Mann et al., 2010) |
| Aerosol activation | ARG (DIAG) | ARG (DIAG) | ARG (PROG) |
| Sub-grid cloud | PC2 (Wilson et al., 2008) | Smith (1990) | Smith (1990) |

Our global model setup is similar to that used by Gordon et al. (2018) and identical to that used in the intercomparison study of Shinozuka et al. (2019). It follows the GA7.1 configuration (Walters et al., 2019), which is the global climate configuration submitted to CMIP6, labelled HadGEM3-GC3.1. The horizontal resolution is $0.8° \times 0.55°$ (N216) and there are 70 vertical levels from the surface to $85\,km$ altitude. The horizontal winds are nudged to ERA-interim starting at $1700\,m$ altitude and ramping up to full strength over the next $453\,m$ of altitude. The relaxation timescale is set to the frequency of the ERA reanalysis files, six hours. The nudging method follows that of Telford et al. (2008). The global model is initialized from the model for August 2016 used by Gordon et al. (2018) and run through to 2017.

This global model drives a $7\,km$-resolution regional simulation centred at $-16°$ latitude, $0°$ longitude, with 670 longitude grid-boxes and 450 latitude grid-boxes, and 140 vertical levels from the surface to $40\,km$ altitude. This simulation is initialized on 17 August at 0000UTC from the global model, and run until the end of 19 August. We use the RA1 (Bush et al., 2019) configuration of the UM with some settings borrowed from the GA7.1 configuration, and UM version 10.8. During the three-day simulation, air masses in the boundary layer advect from approximately the south-eastern corner to the north-western corner of the model domain. Both the global and $7\,km$-resolution models use the single-moment cloud microphysics scheme of Wilson and Ballard (1999) and the double-moment GLOMAP aerosol microphysics scheme (Mann et al., 2010). GLOMAP stands for GLobal Model of Aerosol Processes. Because the aerosol microphysics is double-moment, cloud droplet number

concentration can be represented to some extent in the cloud microphysics and radiation schemes, but as a diagnostic variable rather than a prognostic. We use 'diagnostic' here to indicate that the diagnostic droplet number concentration is calculated from the simulated aerosols, updraft speed, and temperature on each timestep without reference to the droplet concentration on the previous timestep, while the prognostic droplet concentration is retained in memory from one timestep to the next and advected by the simulated wind fields, though it may also be updated if the simulated aerosol concentration or updraft speed changes. The details are explained in the next Section.

The 500 m-resolution simulation uses CASIM double-moment cloud microphysics (Shipway and Hill, 2012) as well as GLOMAP aerosol microphysics. This simulation is driven by the 7 km-resolution model, and has 450 grid-boxes by 450 grid-boxes in latitude and longitude, and the same 140 vertical levels as the 7 km-resolution simulation. It is centred on $-11°$ latitude, $-14.5°$ longitude. Like the 7 km-resolution simulation, we use the RA1 configuration, but a more recent UM version, 11.3 is used, as this has the latest iteration of the CASIM microphysics. Other differences compared to version 10.8 are expected to have only minor effects. This simulation is initialized from the global model on 18 August at 0000UTC. By 19 August, all of the air masses that advect into the domain from the boundaries will have been simulated by the 7 km-resolution model rather than the global model for at least two days, and therefore the resolved wet scavenging processes evident in Figure 2 will have affected the aerosol concentrations, and the higher resolution will have had time to affect the winds. In the domain averages we present, we exclude the 20 grid-boxes nearest to the domain boundaries to remove the transition region between the 7 km and 500 m resolution simulations. The number 20 is arbitrary and chosen by eye, and corresponds to around 30 minutes of advection time for a wind speed of $5\,\mathrm{ms}^{-1}$: enough time to produce some more resolved turbulence but not enough time for full mixing of the boundary layer. We are able to run 500 m-resolution simulations driven by the 65 km global model without the intermediate-resolution nest, but then we would need to exclude more grid boxes at the boundaries of the innermost simulation to allow the high resolution structure to spin up.

The boundary layer scheme we use in all three models is based on that of Lock et al. (2000), which is blended with the Smagorinsky-type scheme from the Met Office Large Eddy Model (Brown, 1999) as grid spacing decreases, as described by Boutle et al. (2014). The scheme is expected to be dominated by the Lock et al. (2000) scheme rather than the Smagorinksy-type scheme, even at the 500 m spatial resolution, while their contributions would be approximately equal at 250 m resolution.

Following the GA7.1 and RA1 configurations of the UM, the global model uses the PC2 subgrid cloud scheme of Wilson et al. (2008) while the regional models use the subgrid cloud scheme of Smith (1990). All three models employ the area adjustment approach of Boutle and Morcrette (2010), so the cloud fraction seen by the microphysics is the mean of that in three sub-layers of each vertical level in each grid cell, while that seen by the radiation code is the maximum.

In the global model and in the higher-resolution regional models, anthropogenic and natural aerosol emissions are taken from the CMIP5 database (Lamarque et al., 2010), except for biomass burning emissions, which are from the Fire Energetics and Emissions Research (FEER) inventory (Ichoku and Ellison, 2014) for August 2017. In addition, sea spray and dust emissions are represented interactively using the parameterizations of Gong (2003) and Woodward (2001) respectively. The offline-oxidants configuration of United Kingdom Chemistry and Aerosols (UKCA) chemistry is used together with dust from the CLASSIC aerosol scheme, as in the HadGEM3-GC3.1 climate model (Mulcahy et al., 2020).

Simulations at all resolutions are fully coupled to the standard radiative transfer scheme in the UM, via the RADAER module for the aerosols (Bellouin et al., 2013). Direct and semi-direct effects of the absorbing aerosols on the cloud are therefore included in the simulations but will be fairly small in this period due to the relatively low aerosol concentrations, and are not the focus of this work.

To illustrate how updraft speeds in clouds are resolved by simulations with different grid sizes, we additionally perform sensitivity simulations at 200 m, 1.5 km and 3 km horizontal resolution, all driven by the 7 km resolution model. These simulations are centered on the same location as the 500 m-resolution simulation, and have the same configuration, with three exceptions. In the case of the simulation at 200 m-resolution, we switch off the sub-grid cloud fraction scheme, and assume clouds are fully resolved by the model. In the case of the simulations at 1.5 km and 3 km resolution, we use 224 grid cells in the horizontal directions, instead of the 450 we use for the 200 m and 500 m simulations, to save CPU time. Lastly, the timesteps in the simulations were adjusted from the 20 s used in the 500 m model to 15 s for the 200 m model, 60 s for the 1500 m model, and 120 s for the 3000 m model.

Our setup can be viewed as an update of that documented by Gordon et al. (2018). However, as the 500 m-resolution model configuration is different to that published by Gordon et al. (2018) (for example, most aerosol-related settings are upgraded to GA7.1 from GA6.1), many of the tunings used in our previous simulations are no longer required. In the wet scavenging code, Gordon et al. (2018) changed a parameter designed to represent the fraction of the area of a grid box over which rain occurs from 30% to 100%. We reverse this change because while it is still more likely that entire 500 m grid-boxes are raining than entire global model grid boxes, we do not account for the evaporation of rain returning scavenged aerosols to the atmosphere. The 30% parameter can therefore be thought of as the fraction of rain which does not evaporate. We have not verified the accuracy of this assumption, which will clearly depend on the regime studied and should be revisited in future. We no longer tune the dry deposition velocity. The biomass burning emissions diameter (specifically, the number geometric mean diameter) is still 120 nm instead of the default for GLOMAP of 150 nm. This diameter is shown by Shinozuka et al. (2019) to give aerosol dry diameters in reasonable agreement (within 40%) with measurements from the parallel NASA ORACLES campaign (ObseRvations of Aerosols above CLouds and their intEractionS), although the diameter is still slightly overestimated compared to ORACLES. For example, in the lower free troposphere most affected by smoke, the simulated dry diameters are biased 37% high. The mass of organic carbon in the same location, which dominates the overall aerosol mass, is biased high by 8%.

## 4 Activation, microphysics and coupling

In the global and 7 km models, a diagnostic activation scheme based on the parameterization of Abdul-Razzak and Ghan (2000) is used to calculate the droplet number concentration (West et al., 2014). We refer to this as the ARG (DIAG) scheme later in the paper. On each timestep, the scheme calculates the droplet concentration at cloud base, and imposes it on grid boxes that are above cloud base and still in the same cloud. The droplet concentration does not depend on the concentration

on previous timesteps. These diagnostic droplet concentrations are used to calculate autoconversion rates in the single-moment cloud microphysics scheme of Wilson and Ballard (1999).

In our $500\,\mathrm{m}$-resolution simulations, aerosols activate in the CASIM code to form cloud droplets, also using the 'ARG' parameterization of Abdul-Razzak and Ghan (2000). Other examples of the use of this parameterization in models at or near the cloud-resolving scale are documented by Ghan et al. (2011), Table 3. The treatment, subsequently referred to as the 'ARG (PROG) activation scheme' where PROG refers to the prognostic droplet concentration, is called once per timestep in grid-boxes when a non-zero mass of water is condensing from the vapor phase. The updraft speed used is set equal to the grid-box mean updraft speed or $0.001\,\mathrm{ms}^{-1}$, whichever is higher. We reduce this threshold from the original $0.1\,\mathrm{ms}^{-1}$ in this paper, which avoids an unphysical spike in the distribution of droplets (Supplementary Figure S1) and an underestimation of the frequency of low droplet concentrations. Instead of the tuned values used by Gordon et al. (2018), the hygroscopicities used are now those recommended by Petters and Kreidenweis (2007) including a kappa value for organic carbon of 0.2. If the number of droplets activated in a timestep exceeds the number of droplets already existing in that grid cell, the droplet concentration used in the microphysics and radiation schemes is updated to the new value. If, on the other hand, the cloud fraction in the grid box goes down, the droplet concentration is altered in proportion, as discussed later. Cloud droplet number concentrations in our $500\,\mathrm{m}$-resolution simulation are also calculated diagnostically on each timestep by the ARG (DIAG) scheme (West et al., 2014), using the same procedure as in the global model, described above. These number concentrations are not used by the model's microphysics or radiation scheme, and so in future simulations this parameterization could be switched off. However, for this study we leave it switched on in order to examine its scale invariance and to compare the predicted droplet concentrations with those from ARG (PROG).

In the sub-grid cloud parameterization, described by Smith (1990), condensation of water vapor onto cloud droplets is treated with 'saturation adjustment' so supersaturation is not prognostic and droplets are assumed to be in equilibrium at the end of each model timestep. We note that because we use a sub-grid cloud scheme, the grid-box mean equilibrium relative humidity in clouds may be below 100%. In the CASIM microphysics scheme, autoconversion and accretion are handled by the parameterization of Khairoutdinov and Kogan (2000). In our simulations the clouds are entirely warm phase; for a description of the representation of cold clouds in CASIM see Miltenberger et al. (2018). In addition to our reduction of the minimum updraft speed used in the activation scheme, we also change the cloud droplet size distribution assumed by the bulk scheme from an exponential distribution to a gamma distribution. This change is explained further in the context of the evaluation in Section 6.7.

The coupling from the GLOMAP aerosol microphysics code to the CASIM cloud microphysics proceeds simply by passing the aerosol mass and number in the soluble Aitken, accumulation and coarse modes, and the volume-weighted kappa values of these modes, to the activation scheme in CASIM. The kappa values are parameters which describe the hygroscopicity of an aerosol chemical component (Petters and Kreidenweis, 2007). The CASIM microphysical process rates are coupled to GLOMAP aerosols following the procedure used in the default configuration of the UM with single-moment microphysics from Wilson and Ballard (1999). The autoconversion and accretion rates are summed and passed back to the aerosol micro-physics code to determine the rate of removal of aerosols in droplets by rain, while the rain and snow rates are used to determine

the rate of impaction scavenging of aerosol by precipitation. Autoconversion and accretion also reduce the prognostic droplet number concentration. The liquid water content is used in the calculation of the rate of conversion of sulfur dioxide to aerosol-phase sulfate inside cloud droplets in the GLOMAP module of the code. The CASIM microphysics code has the capability to simulate cloud microphysical processing of aerosol (Miltenberger et al., 2018), for example the reduction of aerosol number

concentration when cloud droplets collide and coalesce, or the increase in aerosol number concentration when rain evaporates. However, there is no capability to track the composition of the aerosol inside hydrometeors during processing, nor to perform aqueous chemistry in the CASIM module. For simplicity and to save on computational expense, therefore, we do not keep track of aerosols in hydrometeors separately to aerosols outside them, and we keep the aqueous chemistry in the GLOMAP code. Aerosols that activate to form cloud droplets are only removed if they are wet scavenged (i.e. they form rain) and if this

happens it is irreversible: the evaporation of rain does not return aerosols to the atmosphere. A coupled GLOMAP-CASIM double-moment model that includes cloud microphysical and chemical processing of aerosol is deferred to future work.

## 5   Developments to the activation scheme

The procedure for aerosol activation adopted in both ARG (DIAG) and ARG (PROG) activation schemes is to activate aerosols on each timestep as if no cloud were present whenever there is a tendency for water mass to condense. In ARG (DIAG), the

number of droplets that exist in the box before the activation scheme is run is not stored, so the new value of the droplet concentration is used regardless of the previous concentrations. In ARG (PROG), the number of droplets already in the grid-box is stored for the double-moment CASIM microphysics, and if the new number exceeds the old, the number of droplets is increased to the new value. The overwriting of old droplet concentrations by new droplet concentrations if they exceed the old is the procedure of Stevens et al. (1996); Lohmann (2002), and others, but Lohmann (2002) only activated aerosols at

cloud base, and assumed cloud droplet concentrations were uniform in columns within a cloud, a procedure since followed by the ARG (DIAG) activation scheme (West et al., 2014). The procedure in CASIM, as in models such as CAM5.0 that employ the Morrison and Gettelman (2008) activation scheme or its successors, is to do activation via ARG (PROG) at all vertical levels within the cloud. In CASIM, the maximum supersaturation is diagnosed assuming no hydrometeors are present on each timestep, while above cloud base in CAM5.0, a fixed supersaturation of 0.3% is assumed (Wang et al., 2013). The use

of the grid-scale mean updraft in the ARG (PROG) activation scheme, as in RAMS (Saleeby and Cotton, 2004; Thompson, 2016), differs from the scheme of Morrison and Gettelman (2008), which was written with climate model resolution in mind and therefore uses a turbulent sub-grid-scale updraft (Morrison et al., 2005) instead of the grid-scale mean, which is mostly unresolved at low resolution.

There are two possible mechanisms by which the ARG (PROG) scheme for the double-moment CASIM microphysics may

overestimate the impact of in-cloud activation, leading to overestimated cloud droplet concentration. First, the effect of existing cloud droplets on supersaturation is neglected by default, but activation is still repeated every timestep at all levels in the cloud. Second, provided a cloud does not evaporate, the cloud droplet number produced depends on the maximum updraft speed over

the cloud's lifetime rather than the mean updraft speed, though it is not clear that this mechanism leads to an overestimate, or is correct.

Existing cloud droplets certainly affect the supersaturation of water vapor in the cloud. In models like ours with saturation adjustment, there is an assumption that the concentrations of water vapor and liquid water reach equilibrium over one model timestep, which in our case is 20 s. In pre-existing clouds with sufficiently high liquid water content and droplet number, we may assume the sink of water vapor to unactivated aerosols will be negligible, and therefore we may write, after Squires (1952) and others, for example Politovich and Cooper (1988); Korolev and Mazin (2003), an equation for the time evolution of the supersaturation $s$, using the notation of Ghan et al. (2011):

$$\frac{ds}{dt} = \alpha(T)w - \gamma^* G N \bar{r} s \tag{1}$$

Here $w$ is updraft velocity, $\alpha(T)$ is a thermodynamic term that relates the updraft to the tendency for water vapor to condense as it cools, $N$ is the droplet number, $\bar{r}$ the number mean droplet radius, $\gamma^*$ is another term which follows from the thermodynamics of rising moist air with assumptions detailed in Chapter 12 of Pruppacher and Klett (1997), and $G$ is the growth coefficient, which depends on the diffusivity of water vapor in air and on the thermal conductivity of the air. The prescription in Equation 1 is valid only for warm-phase clouds; Korolev and Mazin (2003) describe a more general approach for mixed phase clouds. We correct the diffusivity following the size-independent formulation of Fountoukis and Nenes (2005) except with an accommodation coefficient of 1 as recommended by Laaksonen et al. (2005). If (hypothetically) the system were not in equilibrium and $w$, $T$, $N$ and $\bar{r}$ were constant in time, the supersaturation $s(t)$ could then be approximated by (e.g. Grabowski and Wang, 2013)

$$s(t) = s_{eq} + (s_0 - s_{eq}) \exp\left(-\gamma^* G N \bar{r}(t - t_0)\right) \tag{2}$$

where $s = s_0$ at $t = t_0$. In this equation, $(\gamma^* G N \bar{r})^{-1}$ may be interpreted as $\tau$, the relaxation timescale for the supersaturation, and, in liquid clouds with sufficiently high water content, the supersaturation relevant to aerosol activation is given by the equilibrium or 'quasi-steady' value (Politovich and Cooper, 1988)

$$s_{eq} = \alpha w \tau. \tag{3}$$

Therefore the concentration of newly activated aerosol in each aerosol mode $i$, $N_{d,i,new}$, is related to the concentration of aerosol in that mode $N_{a,i}$ in the same way as in the activation parameterizations, by

$$N_{d,i,new} = \frac{1}{2} N_{a,i} \left( 1 + \text{Erf}\left( \frac{2 \ln \frac{s_{eq}}{s_{c,i}}}{3\sqrt{2}\ln(\sigma_i)} \right) \right) \tag{4}$$

where $\sigma_i$ is the mode width and

$$s_{c,i} = r_{a,i}^{-1.5} \sqrt{\frac{4A^3}{27B_i}} \tag{5}$$

is the critical supersaturation. The coefficient $A$ is a function of temperature and $B_i$ of the particle hygroscopicity, $r_{a,i}$ is the geometric mean radius of the $i^{th}$ aerosol mode, and the equation assumes the aerosols are internally mixed (Pruppacher

and Klett, 1997). The effect on the supersaturation of the condensation or evaporation of rain is currently neglected in these simulations. While the rain water mass is non-negligible compared to the cloud liquid water mass, above cloud base (where it matters) the product of the rain number and the radius is less than 1% of the product of the cloud number and cloud radius at least in the clouds we study here, so the effect on the relaxation time is negligible. At cloud base, the rain mass concentration can exceed the cloud droplet mass, but the small number of rain droplets still means the effect of rain on the relaxation time is negligible.

In well-established clouds with high liquid water content, Korolev (1995); Ming et al. (2007); Ghan et al. (2011) suggested that using Equation 3 should be a better approximation for the supersaturation than the maximum supersaturation $s_{max}$ generated by activation parameterizations such as ARG. Dearden (2009) tested this approximation in large eddy simulations, and found the maximum supersaturation $s_{eq}$ calculated assuming equilibrium with existing droplets to be a much better approximation than that derived using a precursor to the ARG parameterization (Twomey, 1959), which is valid in the approximately same conditions as ARG: at cloud base. In WRF-chem, Yang et al. (2015) found that implementing the suggestion improved simulated wet scavenging. We use the quasi-steady state equation only when it produces a lower supersaturation than the ARG parameterization. The more detailed microphysics scheme of Phillips et al. (2007) also uses a diagnostic parameterization similar to that of ARG at cloud base and a different approach above. However, inside clouds Phillips et al. (2007) represent supersaturation prognostically, without saturation adjustment, in contrast to our cruder quasi-steady approximation.

In a general circulation or mesoscale model with a sub-grid cloud fraction scheme, partially cloudy grid boxes must be accounted for. By default in ARG (PROG) within CASIM, the grid-box mean change in cloud droplet number concentration on each timestep is

$$\frac{\Delta \bar{N}_{d,t+1}}{\Delta t} = \frac{F_{t+1} \times \left( \max \left( (\tilde{N}_{d,t+1} - \tilde{N}_{d,t} F_t / F_{t+1}), 0 \right) \right)}{\Delta t} \quad (6)$$

where $\Delta t$ is the model timestep, 20 s in our tests, $\tilde{N}_{d,t+1}$ denotes the newly calculated in-cloud (denoted by tilde) cloud droplet number concentration, $\tilde{N}_{d,t}$ that calculated on the previous timestep, $\bar{N}_{d,t+1}$ denotes the newly calculated grid-box mean (denoted by overbar) cloud droplet number concentration, and $F_{t+1}$ is the current cloud fraction. The $\tilde{N}_d$ is calculated by running the ARG parameterization over the whole grid box on the assumption that it is completely cloudy so the real grid-box mean cloud droplet number concentration is this value multiplied by the cloud fraction. In some models $\Delta t$ is set to an activation timescale rather than the model timestep (20 minutes in the case of Morrison and Gettelman (2008)). The multiplication by cloud fraction is done after taking the maximum rather than before to handle the case where the cloud fraction increases but the cloud droplet number decreases. The activation scheme is not run when the cloud fraction decreases. When a cloud evaporates, there is a homogeneous mixing assumption: the cloud is assumed to evaporate uniformly across the grid box so that all of the cloud droplets get smaller as the liquid water content decreases, and they are only removed when the cloud fraction or mass of liquid water reach thresholds close to zero ($10^{-10}\,\mathrm{kg\,kg^{-1}}$ for liquid water content and $10^{-12}$ for cloud fraction).

In our improved activation scheme, we assume that in each partially cloudy grid box, the total number of droplets that will either be activated, or remain activated, on a new timestep, denoted by $t+1$, is the sum of those activated inside the old cloud

**Table 2.** Equations for activation to apply in different possible situations as described in the text

| Condition | Equation | Application |
|---|---|---|
| $F_{t+1} - F_t > 0$, $F_{t+1} > 0.05$, and $S_{cloud} < S_{arg}$ | Eq. 8 | Activation inside existing clouds |
| $F_{t+1} - F_t > 0$, $F_{t+1} < 0.05$ or $S_{cloud} > S_{arg}$ | Eq. 6 | Activation in newly forming clouds |
| $F_{t+1} - F_t < 0$ | Eq. 9 | Evaporating clouds |

and those activated in any new cloud that forms. Therefore we replace $\tilde{N}_{d,t+1}$ in Equation 6 by

$$\frac{\tilde{N}_{d,cloud,t+1}F_t + (F_{t+1} - F_t)\tilde{N}_{d,ARG,t+1}}{F_{t+1}}. \tag{7}$$

The first term in this Equation 7 represents aerosols that will be activated at equilibrium in the existing cloud. This term comprises both aerosols that are already contained in large cloud droplets and aerosols that may be newly activated in the

5 existing cloud due to increasing updraft speeds and supersaturations ('secondary activation'). The second term in Equation 7 represents aerosols that will be activated in any additional new cloud that forms in the grid box. We thus obtain

$$\frac{\Delta \bar{N}_{d,t+1}}{\Delta t} = \frac{F_{t+1} \times \left( \max\left( \frac{\tilde{N}_{d,cloud,t+1}F_t + (F_{t+1} - F_t)\tilde{N}_{d,ARG,t+1}}{F_{t+1}} - \frac{\tilde{N}_{d,t}F_t}{F_{t+1}}, 0 \right) \right)}{\Delta t} \tag{8}$$

where $F_t$ is the cloud fraction calculated at the end of the previous timestep and then advected, $\tilde{N}_{d,cloud,t+1}$ is the in-cloud droplet number calculated using Eq. 3 for the supersaturation and $\tilde{N}_{d,ARG}$ is the result of the ARG parameterization (or a

10 similar treatment such as that of Twomey (1959) or Nenes and Seinfeld (2003). In our model, $F_t$ is available even when the sub-grid cloud scheme is diagnostic rather than prognostic, and the equation is not used if $F_{t+1} - F_t < 0$: if this is the case we use instead

$$\frac{\Delta \bar{N}_{d,t+1}}{\Delta t} = \frac{(F_{t+1} - F_t)\tilde{N}_{d,t}}{\Delta t}. \tag{9}$$

If the cloud fraction is below 5% or the in-cloud supersaturation calculated from the cloud droplets via Eq. 3 is higher than that

calculated with the ARG parameterization, we revert to Equation 6, using only the ARG scheme. The equations for the various conditions are summarised in Table 2.

With all of the double-moment approaches we are aware of, over the lifetime of a cloud, the in-cloud droplet number is more likely to increase than it is to decrease, because the droplet number in each gridbox is overwritten if the new droplet number exceeds the old. Therefore, the fraction of activated aerosol will end up corresponding to the highest updraft speed seen during

that lifetime. Morrison et al. (2005) divided the number of additional new droplets activated by 2 to help compensate for this. This 'ratcheting' mechanism is not necessarily unrealistic, since turbulence-induced upward fluctuations in updraft speed in

clouds may well activate more droplets, while downward fluctuations around a positive mean updraft are unlikely to lead to large existing droplets completely evaporating. However, the ratcheting means the double-moment scheme should produce more droplets on average than a single-moment scheme provided the number concentration of droplets is diagnosed from the aerosol concentration in the single-moment scheme. For example, if the ARG (DIAG) activation scheme were fed by the same updraft speeds and the same aerosol as a double-moment scheme, fewer droplets on average would be produced. The droplet concentrations shown later, in Figure 8, suggest that indeed the ARG (PROG) scheme does produce more droplets, but the comparison is complicated by the ARG (DIAG) procedure of setting vertically constant droplet concentrations above cloud base.

In the version of CASIM we are using, aerosol processing is not included, so aerosols are not removed from the gas phase or tracked separately in the cloud phase. Therefore, in our improved scheme, once a cloud forms and the supersaturation used in the activation scheme is reduced due to the existing droplets, the number of aerosols activated on subsequent timesteps within the cloud will be smaller than the number activated the first time the cloud formed. Because the droplet number is only updated if it is higher than the previous droplet number, this will not generally cause the droplet number to decrease, but it will artificially hinder secondary activation from increasing the droplet number. In other words, while our improvements are designed to reduce activation above cloud base, because of the lack of processing we may have reduced it too much. This potential bias could be avoided in future work when aerosol processing is re-introduced to the model.

Aerosol activation is affected by the degree to which updraft speeds in clouds are resolved by a model. If the grid-box mean updraft speed is used in the activation scheme and not all updrafts are resolved, one might expect an underestimate in the concentrations of activated aerosols to result. This bias may counteract the effect of overestimating in-cloud activation discussed earlier. The result of the study of Malavelle et al. (2014) is that the ratio of the variance of simulated updrafts to the variance of real updrafts is given by

$$R = 1 - \frac{\left(\frac{\Delta x}{Z_{ml}}\right)^{E1} + a\left(\frac{\Delta x}{Z_{ml}}\right)^{E2}}{\left(\frac{\Delta x}{Z_{ml}}\right)^{E1} + b\left(\frac{\Delta x}{Z_{ml}}\right)^{E2} + c} \tag{10}$$

for grid resolution $\Delta x$, a parameter $Z_{ml}$ proportional to the boundary layer height as described in Supplementary Table S4, and fitted constants $E1 = 2.59$, $E2 = 1.34$, $a = 7.95$, $b = 8.00$ and $c = 1.05$. An additional correction factor $f$ is required to account for the difference between effective and actual grid resolution. This factor varies from model to model but for the UM it is a factor four correction to the variance (Malavelle et al., 2014). The constant of proportionality between $Z_{ml}$ and the boundary layer height depends on the cloud regime. The updraft velocity in the activation scheme can then be set to

$$w_{act} = w\sqrt{\frac{f}{R}} \tag{11}$$

to correct for unresolved fluctuations in the velocity distribution, following Equation 21 in Malavelle et al. (2014).

In the cloudy part of the model domain, the mixed layer height is around $2000\,\mathrm{m}$, so the formulation, accounting for effective grid resolution, suggests a scaling factor of approximately 3 (see Supplementary Table S4). The comparison of the PDF of updraft speeds between the model and observations suggests a factor that varies from 1.5 to 4.2 depending on altitude

and liquid water content. We give more weight to results at cloud base and to the data binned by liquid water content in Supplementary Figure S6, so we use a factor of 2. We do not implement the Malavelle et al. (2014) scheme in full as it appears to exaggerate the scaling factor required for the model resolutions and boundary layer we are studying. Moreover, we note that scaling all our updrafts up by a factor two doubles the mean updraft speed used in the activation scheme as well as the standard deviation. For the cloud we simulate, at cloud base and at cloud top this leads to poorer agreement of the mean updraft speed with observations, while in the middle of the cloud the agreement improves. The potential for such a scaling to introduce bias requires further study.

## 6 Evaluation of the double-moment microphysics

In this section we evaluate the configuration of the model with double-moment microphysics without the additional developments described in the previous section, to establish a baseline for testing these additional developments. All of our evaluation is performed with a snapshot of the model at 1200 UTC on 19 August. The aircraft took off at 1001UTC, entered the domain of the 500 m model at 1103 UTC, left it at 1310 UTC and landed at 1344 UTC. We assume that changes in meteorological or aerosol variables in the region between 1103 and 1310 UTC are small.

### 6.1 Satellite liquid water path and cloud droplet number

Figure 3 shows the liquid water path and cloud droplet number in the simulation domain compared to the MODIS instrument on the TERRA satellite (Platnick et al., 2015). The calculation of droplet number concentration from MODIS effective radius, cloud top height and optical depth is described by Gordon et al. (2018) and follows Boers et al. (2006) and Grosvenor and Wood (2014). The cloud feature is reproduced by the model, but it is substantially smaller than that observed by the satellite and it is displaced to the south-east. The south-easterly displacement suggests the wind speed in the boundary layer in the 7 km model is likely a little slower than in reality. The liquid water path seems to match the satellite well in the 7 km simulation. In the 500 m simulation, the in-cloud liquid water path is also realistic, but the relative frequency of intermediate liquid water paths between 100 and 300 $\mathrm{g m^{-2}}$ is underestimated compared to the satellite. We were able to increase the cloud cover in the 500 m-resolution model by tuning the critical minimum relative humidity in the sub-grid cloud scheme, as shown in Supplementary Figure S2. However, to maintain consistency with the official RA1 UM configuration (Bush et al., 2019) we retain the default values of these parameters in the simulations that follow.

The retrieval of cloud droplet number does not yield data in all cloudy pixels due to confusion with overlying cirrus cloud which prevents the retrieval of cloud top temperature. However, from the valid pixels, the cloud droplet concentration in the 7 km simulation is realistic. In the 500 m simulation, a large number of isolated high peaks in the droplet concentration are seen, while there are no such high peaks in the measurements. In the simulation, these peaks correspond to high updraft speeds at some point in the history of the cloud. It is possible that these peaks correspond to the few grid cells where our saturation adjustment assumption has led to excessive release of latent heat, temporarily strengthening the updrafts. In practice, this variability may not bias the area average: Supplementary Figure S3 shows the same figure, except with the central subfigures

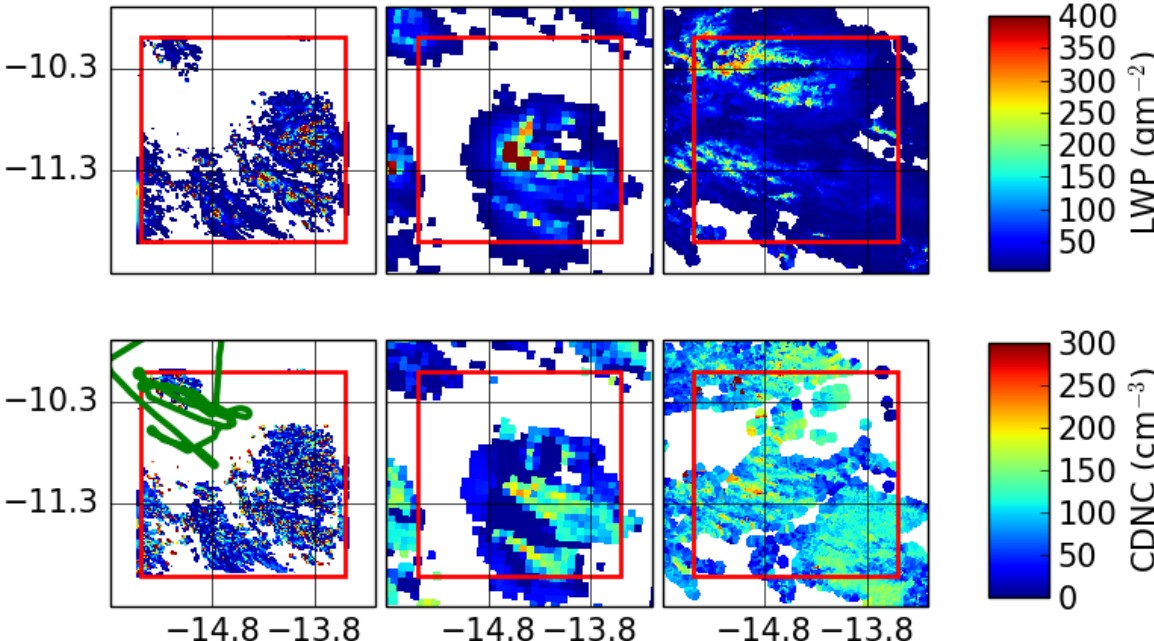

**Figure 3.** Liquid water path (LWP, top) and cloud droplet number concentration (CDNC, below) compared to MODIS on 19 August at midday. Instantaneous output from the 500 m-resolution simulation is shown on the left, the 7 km-resolution simulation in the centre, and MODIS TERRA on the right. The sizes of isolated MODIS pixels are increased during regridding, for ease of viewing. There is not always a valid MODIS CDNC retrieval where there is a valid liquid water path, so the cloud extent can only be inferred from the top-right figure. The green trace on the bottom-left figure shows the path of the aircraft.

replaced by the droplet concentration and water path in the 500 m simulation regridded to 5 km resolution, which is not dissimilar to the apparent resolution of the relatively sparsely sampled MODIS cloud droplet number concentration. The simulations in these subfigures show better agreement with the observations.

## 6.2 Temperature and humidity

5   Figure 4 shows the aircraft measurements of temperature and relative humidity compared to the global and 7 km-resolution simulations in the left and central sub-figures. These measurements are supported by Supplementary Figure S4, which shows the corresponding data from radiosondes at Ascension Island compared to the regional and global simulations. The simulations clearly underestimate the boundary layer height, by around 400 m, with the 7 km simulation performing slightly worse than the 500 m simulation. The grid spacing at the top of the boundary layer is approximately 80 m. This underestimate will affect our

10   comparison of simulated cloud properties, especially close to cloud top, to observations, as we discuss later. The underestimate is partly due to an underestimate of the boundary layer height by the ERA-interim meteorology used to nudge the global

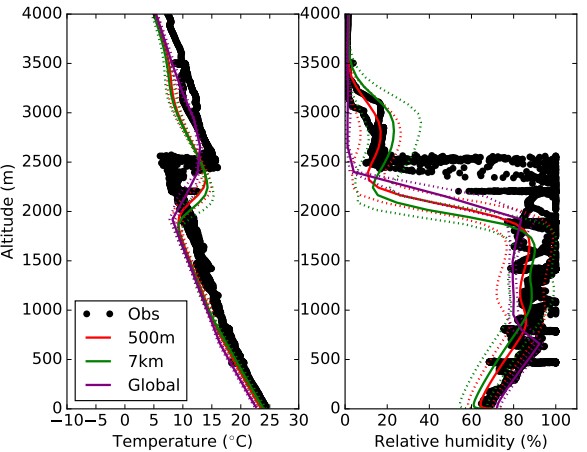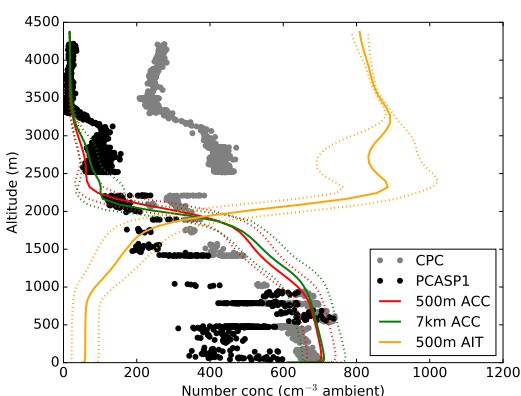

**Figure 4.** FAAM aircraft observations of temperature (left), relative humidity (centre) and out-of-cloud Aitken (AIT) and accumulation (ACC)-mode aerosol number concentration (right) with the corresponding model means over the 500 m domain. The aircraft data are used whenever they are in the domain of the 500 m-resolution simulation. Variability across the model domain is assumed here to be small, but, for accumulation-mode aerosol, it may be inferred from Figure 2. Dotted lines indicate one standard deviation. Aerosol data inside clouds are removed as described in Section 2.

simulation, and it is exacerbated by the 7 km-resolution simulation. We verified that the bias persists in the more up-to-date ERA5 reanalysis. We were able to reduce the underestimate by about 100 m by tuning entrainment parameters in the global model, but as this gain is small relative to the overall 400 m bias, we retain the default configuration in the following analysis. We also do not try to correct for this discrepancy in the evaluation by comparing lower altitudes in the model to higher

altitudes in observations, as making such a correction would lead to further complications from the different temperatures at the different altitudes. The radiosondes from Ascension Island, which lies just outside the domain of the 500 m -resolution simulation, indicate that the boundary layer height there is lower than the boundary layer height inside the 500 m simulation domain, by approximately 500 m in the observations and 300 m in the 7 km-resolution model.

     As well as the error in the boundary layer height, the temperature is generally underestimated by the simulations by around

1.5°C above the boundary layer, although the simulations do reproduce the hint of a secondary inversion at around 3.5 km altitude reasonably well. The relative humidity is slightly underestimated in most of the boundary layer compared to the aircraft data. Unlike the humidity derived from the aircraft, the radiosondes record lower humidities than the model. This discrepancy may be due to imperfect matching between the clouds in the model grid boxes and the observations as we also saw in Figure 3, or due to effects from Ascension Island, which is not included in the model. Above cloud, all simulations produce an elevated

relative humidity in the moderately polluted aerosol layer, in good agreement with the aircraft measurements in Figure 4.

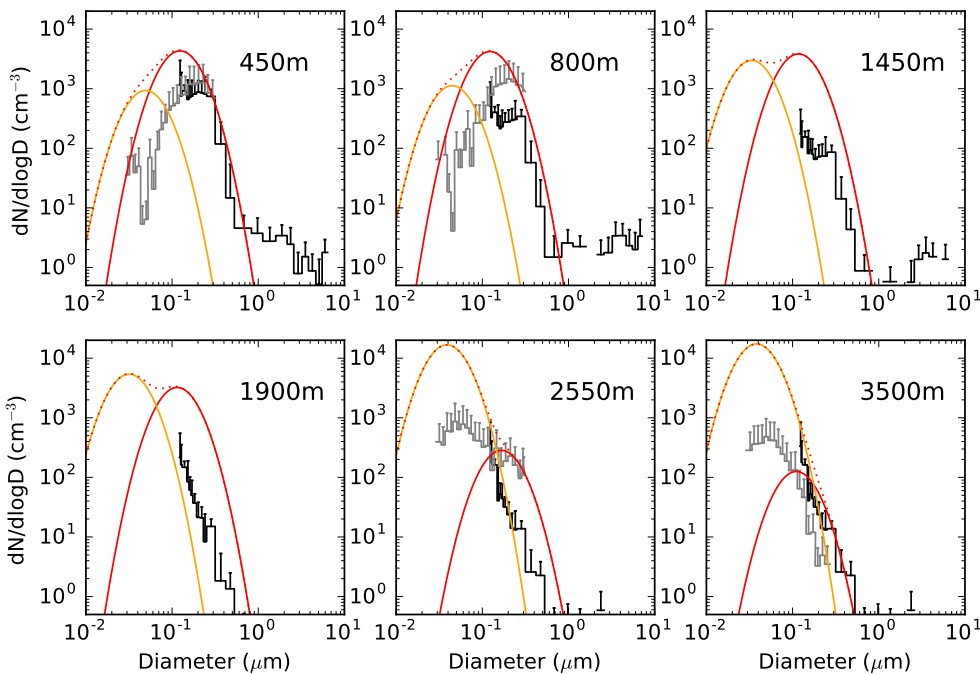

**Figure 5.** The dry size distributions (out of cloud) observed by the aircraft from a combination of PCASP (black) and SMPS (grey) instruments are shown at different altitudes, compared to the model Aitken (orange) and accumulation mode (red) number concentrations, with the total shown as a red dotted line. The simulated coarse mode is not shown. All data shown are out-of-cloud at standard temperature and pressure. The simulated data are a mean over the domain at 1200 UTC. Only aircraft data from straight-and-level legs at the specified altitude are shown.

## 6.3 Aerosol number concentration and size distribution

The simulated vertical profiles of aerosol number concentration are compared to observations on the right of Figure 4. The observed accumulation-mode aerosol number concentration in the boundary layer varies from 350 to $700\,\mathrm{cm}^{-3}$ below $1000\,\mathrm{m}$ altitude, and 200 to $400\,\mathrm{cm}^{-3}$ between 1000 and $2000\,\mathrm{m}$. The domain mean in the simulation, around $700\,\mathrm{cm}^{-3}$ below $1000\,\mathrm{m}$ and $600\,\mathrm{cm}^{-3}$ above, is biased high compared to the observations. The Aitken mode number concentration above the boundary layer is over-predicted by the model by a larger amount, more than a factor of two. The overestimate is due to excessive new particle formation in the upper troposphere in the global simulation. The full vertical profile of Aitken-mode particle concentrations in the global model is shown in Supplementary Figure S5. The bias is also present in the evaluation of Mulcahy et al. (2020). The excessive new particle formation is itself due at least in part to an overestimate of sulfur dioxide concentrations in the model (not shown). The overestimation of the Aitken mode number concentration is likely responsible in part for overestimates in the cloud droplet concentration at the top of the simulated clouds (around $2215\,\mathrm{m}$), as discussed in Section 6.6.

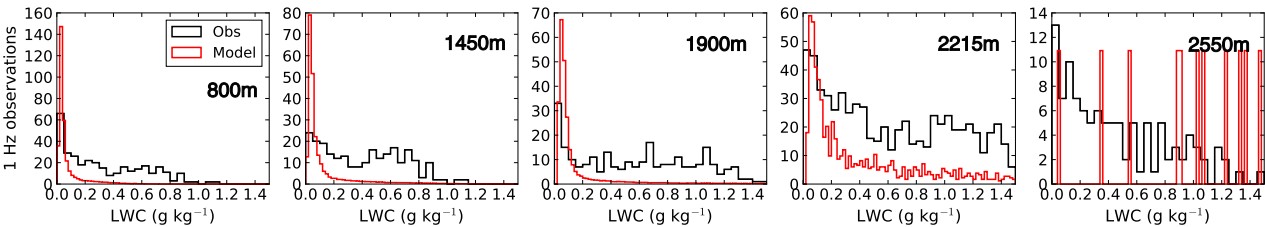

**Figure 6.** Frequencies of observed in-cloud liquid water contents from the cloud droplet probe compared to the liquid water contents in cloudy grid cells in the 500 m- resolution simulation at the five altitudes marked. We include all model grid cells at the specified altitude within the simulation domain with a liquid water content greater than $0.01\,\text{gkg}^{-1}$ at the instant of 1200 UTC on 19 August in the histogram, except for the grid cells within 10 km of the domain boundary. The histograms of the simulated liquid water contents are scaled to contain the same number of entries as the histograms of the observations (so, for example, there are 11 model grid boxes plotted at 2550 m altitude).

It will be important to study the scale invariance of nucleation-mode microphysics at high resolution, and the biases in its parent model, which also affect the UK contributions to the CMIP6 experiments, in future work.

The observed aerosol size distribution is shown in Figure 5. As suggested by the number concentrations in Figure 4, the Aitken mode concentration is consistently overestimated, but the accumulation mode dry diameter is simulated well, generally within 30% of observations, though it appears to be overestimated by a larger amount at the 1900 m level, the altitude of the cloud layer (note that cloudy grid boxes are excluded from the average). The limited amount of aerosol data available at the level of the clouds precludes a more quantitative comparison at this altitude. This overestimation may indicate too much aqueous sulphate production, or too little rainout of the larger particles.

### 6.4 Liquid water content

The aircraft targeted areas of strong radar reflectivity, and so its sampling of liquid water content is biased towards thicker clouds. This bias must be addressed as we compare the observations with the whole range of model grid cells in our domain, as discussed by Field and Furtado (2016). The distribution of liquid water content at the five altitudes sampled by the aircraft is compared to the distribution at the same altitudes of the cloudy grid-boxes of the 500 m- resolution model in Figure 6. Unsurprisingly, the observed liquid water contents are skewed towards high values while the simulated liquid water contents are mostly much lower. To compare vertical velocities and cloud droplet number concentrations more fairly between simulations and observations, for some of our analysis we split the samples at a liquid water content of $0.15\,\text{gkg}^{-1}$. The difference in the distribution of liquid water content between the model and the observations within each of these bins is then much smaller than the difference over the full range of liquid water content (Figure 6). In some of the subsequent evaluation, we focus on observations and simulation data made at liquid water contents above $0.15\,\text{gkg}^{-1}$, as this is where most of the observations lie. The histograms of vertical velocities in two bins of liquid water content shown in Supplementary Figure S7 show that, as expected, high liquid water content is associated with updrafts while low liquid water content is more likely in downdrafts.

## 6.5  Updrafts

We compare simulated grid-box mean updraft speeds in our $7\,\mathrm{km}$ and $500\,\mathrm{m}$-resolution models, and in the $200\,\mathrm{m}$ sensitivity simulation, to observations in Figure 7. We include all grid boxes where the overall cloud liquid water content is greater than $0.01\,\mathrm{gkg}^{-1}$ irrespective of the cloud fraction. Clearly the width of the distribution is underestimated substantially, even in the $200\,\mathrm{m}$ simulation. An underestimate is expected, because the $200\,\mathrm{m}$ and $500\,\mathrm{m}$ model grid boxes do not resolve all the updrafts. On the other hand, the aircraft records vertical velocity at $32\,\mathrm{Hz}$, corresponding to approximately one measurement for every $4\,\mathrm{m}$ it traverses, and therefore it should represent all of the turbulence that would be captured by a large eddy simulation. In our model, we therefore rely on parameterized boundary layer mixing. At $200\,\mathrm{m}$ resolution, the width of the spatial distribution of updraft speeds is slightly wider than at $500\,\mathrm{m}$, (as shown later in Table 4), and the frequency of cloudy grid cells at $2550\,\mathrm{m}$ altitude is increased. The increased cloudiness at high altitude is presumably because the in-cloud updraft speeds are better resolved. However, the domain-averaged boundary layer height in these simulations is almost the same as that at $500\,\mathrm{m}$ resolution. The low bias in the boundary layer height is mainly due to the driving 7km-resolution and global simulations. In the $7\,\mathrm{km}$ simulations, almost no variability in updraft speed in these clouds is resolved.

Model resolution explains only part of the underestimated variability in vertical velocity. The other reason the updraft widths are too narrow is sampling biases: the sampling of clouds with high radar reflectivity discussed in the previous section. We compare the moments of the simulated and observed updraft distribution in bins of cloud liquid water content, in Supplentary Figure S6. The distributions and their moments when split into two bins are shown in Supplementary Figure S7, which also includes the $200\,\mathrm{m}$ resolution simulation, and Supplementary Table S2. When split by liquid water content, the width of the vertical velocity distributions in the model matches observations better, although it is still underestimated. Before splitting by liquid water content, the mean vertical velocity at each altitude was generally within $0.1\,\mathrm{ms}^{-1}$ of zero, but after the split this is no longer the case, as indicated in Supplementary Table S2. We note that the two variables are not independent: large positive liquid water contents are correlated to large positive vertical velocities because high vertical velocities imply high supersaturations. The reader is also reminded that the $2215\,\mathrm{m}$ altitude level is probably more representative of cloud top in the model than the $2550\,\mathrm{m}$ level, due to the underestimated height of the boundary layer.

If we approximately correct for the sampling bias associated with the path of the aircraft by only considering vertical velocities where liquid water content exceeds $0.15\,\mathrm{gkg}^{-1}$ and also smooth the observed vertical velocity distribution, to show the mean observed vertical velocity over approximately $500\,\mathrm{m}$, we would expect good agreement of the model with observations. When we do this, as shown in Supplementary Figure S8, we do see substantially improved agreement of the model with observations at cloud base and cloud top, but not in the middle of the cloud: the model still underestimates the variability in updraft speed. The residual bias may be a sampling artefact we did not successfully remove, or it may indicate that there is not enough simulated convection.

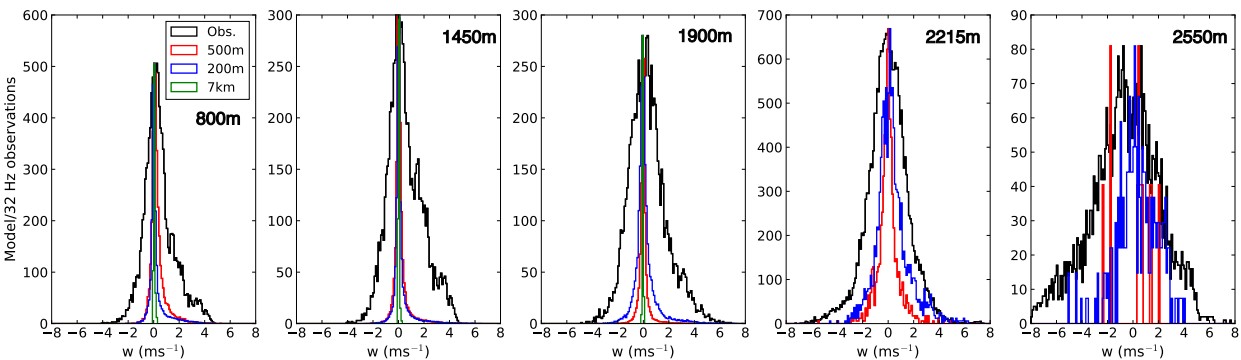

**Figure 7.** Updraft speed in clouds in 32 Hz CLARIFY observations and in the model (before changes to the activation scheme), at altitudes of 800, 1450, 1900, 2215 m and 2550 m from left to right. Model data are from instantaneous simulation output of the 200 m, 500 m and the 7 km-resolution simulations at 1200 UTC on 19 August 2017. Only the area of the 7 km-resolution simulation that overlaps with the 500 m-resolution simulation is included. Data are selected as being in-cloud if the liquid water content exceeds $0.01\,\mathrm{g\,kg^{-1}}$ in both the model and in the observations. The histograms of simulated vertical velocity are scaled so their maxima match the maximum of the histogram of the observed vertical velocity.

## 6.6 Droplet concentrations

The simulated in-cloud droplet number concentration is calculated by first dividing the grid-box mean droplet number concentration by the cloud fraction, then removing any grid cells where the in-cloud mean liquid water content is below $0.01\,\mathrm{g\,kg^{-1}}$ or where the cloud fraction is below 0.05. With the ARG (PROG) activation scheme, Table 3 shows that the mean droplet

concentration is overestimated by a factor of two to three depending on altitude in the clouds with higher liquid water content, and by a larger factor in clouds with low liquid water content. The overestimate is likely to be due in part to the substantial overestimate in aerosol concentrations in the accumulation mode, and in part due to the activation scheme and warm rain representation (discussed later). Figure 8 shows the spatial distributions of the simulated and observed droplet concentrations (using the whole sample, not split by liquid water content). We do not plot droplet concentrations at 2550 m altitude due to

the low sample size. The figure shows the variability (the width of the distribution) is also overestimated, although the frequency of very low droplet concentrations is underestimated. Table 3 shows the overestimate in mean droplet concentration is more severe than the figure suggests, because the aircraft targeted areas of strong radar reflectivity and therefore preferentially sampled high liquid water contents and high droplet concentrations. At 2215 m altitude, it seems likely that the overestimated mean and width of the droplet concentration spatial distribution is at least partly due to activation of the Aitken mode, given

that the number concentration of the Aitken mode is substantially overestimated as discussed in Section 6.3.

The ARG (DIAG) activation scheme in the same simulation also overestimates the variability in droplet concentration, but the mean is closer to observations. Most likely, ARG (DIAG) produces fewer droplets than ARG (PROG) because the ARG (PROG) droplet number depends on the highest updraft in the history of the cloud while the ARG (DIAG) droplet number depends on the updraft at the particular instant the droplet number is diagnosed, as discussed in Section 5. In ARG (DIAG) at

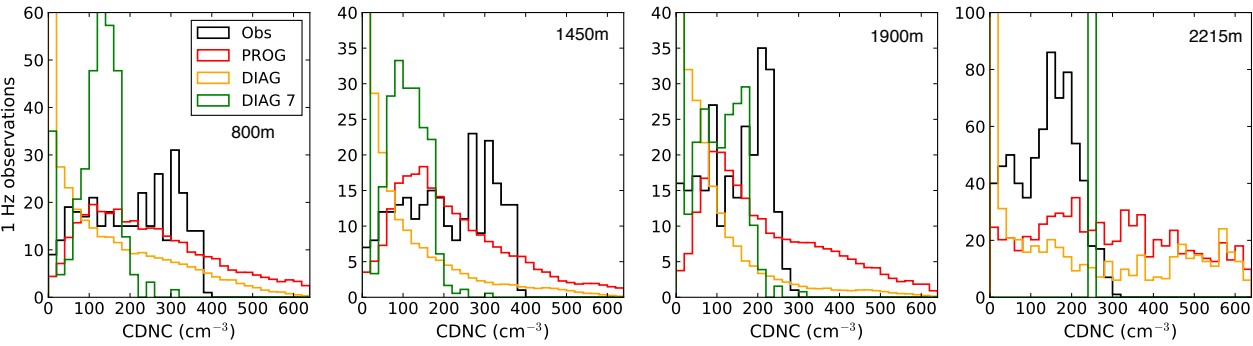

**Figure 8.** Cloud droplet number concentration in CLARIFY observations at 1 Hz, in the 500 m resolution model as predicted by ARG (PROG), labelled 'PROG' in CASIM and by ARG (DIAG) with its updraft PDF, labelled 'DIAG', and in the 7 km resolution model as predicted by ARG (DIAG), labelled 'DIAG 7' at altitudes of 800, 1450, 1900, and 2215 m from left to right. The histograms of simulated cloud droplet number concentration are scaled to contain the same number of entries as the 1 Hz observations. The histograms show droplet numbers over the full range of in-cloud liquid water contents. The observations are compared to gridboxes sampled from the whole model domain, except for the 20 gridboxes nearest the boundaries. Values are grid-box means rather than in-cloud means. Strictly, the aircraft traverses 500 m in about 4 seconds, so 1 Hz is too high a sampling frequency for comparison to the model. However, down-sampling to 0.25 Hz reduces the data sample without changing the shape of the distribution.

7 km resolution, by contrast, the variability is underestimated and the mean droplet concentration is also underestimated by a factor often around two. This ARG (DIAG) activation scheme (so far only used at convection-permitting resolution by Gordon et al. (2018)), also produces a spike in the very lowest bin. This spike is the result of a lack of scale-awareness in the scheme as it is currently coded. It corresponds to the minimum droplet concentration being assigned, $5\,\mathrm{cm^{-3}}$, because the characteristic

updraft speed is out of the range allowed in the code (a more detailed discussion is given in the supplement). Grid-boxes with updrafts out-of-range are seen in the 7 km-resolution simulations as well as the 500 m simulations. We describe this effect in more detail, and propose a fix to this specific problem, in Supplementary Section 12.

The lack of scale invariance in the ARG (DIAG) diagnostic activation scheme also means the 7 km simulation does not yield the same distribution of cloud droplet concentrations as the 500 m simulation. At 1900 m altitude the simulations agree on the

10 domain-mean droplet concentration: it is $86\,\mathrm{cm^{-3}}$ in the 500 m simulation and $92\,\mathrm{cm^{-3}}$ in the 7 km simulation, while at 800 m altitude they do not: the droplet concentration is $181\,\mathrm{cm^{-3}}$ in the 500 m simulation and $117\,\mathrm{cm^{-3}}$ in the 7 km simulation. The reduced variability in the 7 km simulation compared to the 500 m simulation is expected, but further work is needed to ensure the means are consistent. However, because the diagnostic droplet concentrations in the 7 km model do not feed through the lateral boundaries of the 500 m simulations, biased droplet concentrations in our 7 km model should not substantially affect

the 500 m model.

The spike at $5\,\mathrm{cm^{-3}}$ is not observed in the global climate model and, partly because of this, there is no substantial low bias in the mean droplet concentration. Other reasons for the lack of low bias in the climate model could be the lack of resolved wet scavenging of aerosols. A global evaluation of droplet number in the climate model is presented by Mulcahy et al. (2018).

**Table 3.** Cloud droplet number concentrations in all clouds, and in two bins of liquid water content. The 500m-resolution, 7 km-resolution, and global models are shown, with the ARG (DIAG) and ARG (PROG) activation schemes where they are run. In both model and observations, the threshold liquid water content to define a cloud is $0.01\,\mathrm{gkg}^{-1}$. All data is shown above the first horizontal line, the model grid cells and observations with in-cloud liquid water content above $0.15\,\mathrm{gkg}^{-1}$ are shown above the second horizontal line, and those with liquid water content below $0.15\,\mathrm{gkg}^{-1}$ are shown below. The global model, labelled 'Global DIAG' is not separated by liquid water content as the number of grid-boxes in the domain is too small.

| Altitude | Obs (cm$^{-3}$) | 500 m PROG (cm$^{-3}$) | 500 m (cm$^{-3}$) DIAG | 7km DIAG (cm$^{-3}$) | Global DIAG (cm$^{-3}$) |
|---|---|---|---|---|---|
| 800 | 200 | 253 | 144 | 118 | 180 |
| 1450 | 206 | 229 | 88 | 92 | 180 |
| 1900 | 144 | 227 | 79 | 92 | 145 |
| 2215 | 138 | 301 | 192 | 250 | 180 |
| 2550 | 101 | 370 | 364 | - | - |
| 800 | 249 | 512 | 280 | 180 | - |
| 1450 | 247 | 416 | 164 | 139 | - |
| 1900 | 173 | 421 | 131 | 127 | - |
| 2215 | 158 | 383 | 234 | 250 | - |
| 2550 | 126 | 388 | 338 | - | - |
| 800 | 117 | 220 | 127 | 116 | - |
| 1450 | 89 | 183 | 69 | 86 | - |
| 1900 | 45 | 174 | 65 | 78 | - |
| 2215 | 45 | 156 | 118 | - | - |
| 2550 | 14 | 182 | 647 | - | - |

We considered the possible effect of biases in the underlying ARG algorithm on the results of the ARG(PROG) activation scheme by comparing it to a cloud parcel model with an explicit activation scheme following Köhler theory (Köhler, 1936; Petters and Kreidenweis, 2007; Rothenberg and Wang, 2016). Supplementary Figure S9 shows the fraction of accumulation-mode aerosols activated at cloud base as a function of updraft speed from simulations with our model, run at $200\,\mathrm{m}$ resolution with no sub-grid cloud fraction (for ease of interpretation). We ignored possible contributions from the Aitken and coarse modes. Superposed on the figure, we show simulations with the parcel model in red, and the predictions of the ARG algorithm run offline in orange. We used the Pyrcel parcel model of Rothenberg and Wang (2016) to perform the parcel model and offline ARG calculations. We set the accumulation-mode aerosol and thermodynamic parameters to be consistent with those included in, or simulated by, the Unified Model. We assumed the aerosol was composed of ammonium sulfate (kappa= 0.61), and additionally ran the parcel model again using a kappa value of 0.2 instead of 0.61, to see how the results would change if the aerosol was instead organic carbon. The activated fractions from the parcel model and the ARG parametrization agree to within 20% for both hygroscopicities, confirming that the ARG parameterization is appropriate. The fractions in the UM are more scattered because cloud droplet number is prognostic, and can undergo autoconversion, accretion and sedimentation.

## 6.7 Cloud and rain particle size distributions

We compare the cloud and rain size distributions between the model and the observations in Figure 9. We find the model represents large cloud droplets well, but overestimates the number of small droplets and underestimates the number of large rain drops. The cloud droplet size distributions are broadly similar for cloud liquid water contents below and above $0.15\,\mathrm{g\,kg^{-1}}$, as shown in the Supplement as Figure S11. The simulated surface rainfall amount is compared to observations from the GPM satellite dataset (Huffman et al., 2014) in Supplementary Figure S12. Despite the poor representation of the rain size distribution observed by the aircraft, the total surface rainfall rate simulated by the model is in generally good agreement with the satellite data. Simulated vertical profiles of rain mass and number concentration are evaluated in Figure 12, which is shown later as it includes model developments discussed in Section 5. This evaluation confirms that, compared to the aircraft data, the rain mass is generally well simulated but the number concentration of small drops is overestimated and that of very large drops underestimated. We note that the autoconversion and accretion parameterizations from Khairoutdinov and Kogan (2000), or the parameters of the gamma size distribution, could in principle be tuned to attempt to address this bias, but we did not attempt to do so. Such tuning would likely only be applicable to the clouds that were sampled by the aircraft, which, as we have discussed, are not necessarily even a fair sample of the clouds in our model domain as the aircraft was targeting areas of high radar reflectivity (Field and Furtado, 2016).

In the simulations shown in Figure 9, we have used a gamma size distribution instead of the default exponential cloud droplet size distribution assumed by CASIM and published by Grosvenor et al. (2017); Miltenberger et al. (2018) and others. We find (not shown) that once both the moments for droplet mass and number concentration are predicted by the model, adjusting the prescribed size distribution shape (e.g. changing from an exponential to a gamma distribution) without changing these moments does not strongly affect the important results of the microphysics parameterization, such as rain formation rates, in the case we study, though this may not be true in other clouds. However, in Section 5 we calculate the number mean droplet radius in order to determine supersaturation, and this is sensitive to the shape. For example, an exponential size distribution has a number mean radius equal to one-third of the effective radius and 0.55 times the volume mean radius, while the more realistic gamma size distribution with $\mu = 5$ has a number mean radius equal to 75% of the effective radius and the effective radius is larger than the volume mean radius by 15%. This 15% is still larger than the 8% found in observations (Freud et al., 2011), but in much better agreement than the 66% factor by which the effective radius of the exponential distribution is greater than its volume mean radius. The gamma size distribution is given by

$$N(D) = \frac{N_d \lambda^{\mu+1} D^\mu \mathrm{e}^{-\lambda D}}{\Gamma(\mu+1)} \tag{12}$$

for drop diameter $D$, shape parameter $\lambda$, and droplet concentration $N_d$. The size distributions shown in Figure 9 and used in our subsequent simulations have $\mu = 5$. The use of $\mu = 5$ is inspired by the relationship of Martin et al. (1994) used by Morrison and Gettelman (2008), which yields $\mu = 5$ for a cloud droplet number concentration of $240\,\mathrm{cm^{-3}}$, close to the concentrations observed here. The exponential size distribution, the size distribution used by Morrison and Gettelman (2008), and the gamma distribution with $\mu = 5$ are compared in Supplementary Figure S10 and the number mean droplet radii that result are tabulated

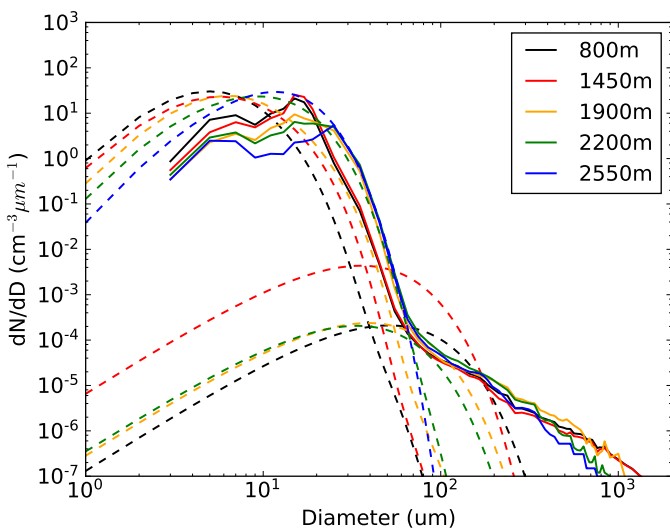

**Figure 9.** Evaluation of cloud and rain droplet size distributions (in microns) at the altitudes sampled by the aircraft. The simulations are from the $500\,\mathrm{m}$-resolution model (the only simulation with double-moment cloud microphysics). The observed size distributions are shown by solid lines and the simulated distributions by dotted lines. The observation data are combined from the cloud droplet probe (CDP), two-dimensional stereo probe (2DS) and cloud imaging probe (CIP). The CDP data and simulated cloud droplet concentrations are filtered to be in-cloud, i.e. to include only samples with $\mathrm{LWC} > 0.01\,\mathrm{g\,kg^{-1}}$, while the CIP data and simulated rain are not.

.

in Supplementary Table S3. The gamma size distribution we use overestimates the number of small droplets compared to observations, but it captures the number of larger cloud droplets very well.

Simulations by Pinsky and Khain (2002) and others showed that in principle a bimodal droplet spectrum could arise as a result of secondary droplet nucleation. We have some indications, from Figure 7, that updrafts may accelerate in our clouds, leading to secondary activation, as strong updrafts, above $4\,\mathrm{m\,s^{-1}}$, are more frequently observed at higher altitudes. It is then tempting to claim the droplet spectrum in Figure 9 is a bimodal distribution with peaks at $7\,\mathrm{\mu m}$ and about $20\,\mathrm{\mu m}$. When we filter the observed size distribution to exclude low liquid water contents in which secondary activation is not expected we find (on the right of Supplementary Figure S11) that at high altitudes in the cloud the slightly bimodal distribution remains. However, it could easily be an instrument artefact, and it could probably also arise from different aerosol types, entrainment, or collision-coalescence processes. Thus it is very speculative to suggest secondary activation is actually taking place. In our simulations, by contrast, there is no mechanism for secondary activation to lead to a bimodal size spectrum: the new droplets must fit into the existing size spectrum definition.

# 7 Results of improvements to the ARG (PROG) activation scheme

For our 500 m simulations, Figure 10 shows that when we correct the ARG (PROG) activation scheme in our model to account for the effect on supersaturation of existing droplets, and the effect of unresolved vertical velocities, we obtain relatively similar cloud droplet numbers to those we started with, with a slight improvement to the agreement of the model with observations. Overall, it looks like the factor 2 correction to the updraft (as suggested by observations at cloud base) produces cloud droplet number in good agreement with the default model for this resolution. However, as well as being resolution-dependent, this finding may be specific to the type of cloud we study, and will likely be influenced by other sources of bias - for example, the assumption that the aerosols are internally mixed, the assumptions about hygroscopicity, or errors in simulating the aerosol size distribution.

The mean maximum supersaturation from the ARG parameterization is around 0.25% at cloud base in the default simulation and around 0.35% when the updraft speed is multiplied by two, in line with expectations (Ghan et al., 2011). The maximum diagnosed supersaturations in new clouds, calculated by the ARG parameterization, and in existing clouds calculated with the quasi-steady-state assumption, are shown as in-cloud domain mean vertical profiles in Figure 10 and as in-cloud histograms in Supplementary Figure S14. They demonstrate that in-cloud activation is clearly not negligible even in this relatively shallow cloud: the mean in-cloud maximum supersaturation can exceed 0.1%. However, in the cloud we study, it is lower than the 0.3% prescribed for convective clouds above cloud base by Wang et al. (2013). The standard deviation of the maximum supersaturations (not shown, for clarity) is also around 0.1% for the case when the updraft speed is multiplied by two, and 0.05% when the updraft is not corrected. The mean in-cloud value is similar to the 0.1% found in a smaller cumulus cloud by Politovich and Cooper (1988). Very occasionally, the relaxation time can exceed the model timestep of 20 s, as shown in the histograms in Supplementary Figure S14, and in this case a bias will result, but the number of grid boxes in which this happens is not significant.

When we account for the effect of existing cloud droplets on supersaturation in the activation scheme, the instantaneous concentration of new droplets activated is always lower above cloud base than when we do not. Increasing the updraft speed increases the activation at cloud base, leading to more activation at cloud base, but, for a factor of two increase in updraft, the number of droplets activated inside the cloud remains lower than in the original model. This difference in vertical profile propagates to the prognostic concentration of cloud droplets, and the consequence is a lower cloud droplet concentration where it is most important for radiative transfer - where the domain-mean liquid water content is highest close to cloud top– compared to the droplet concentration at cloud base.

The spatial distributions of cloud droplet number concentration, shown in Figure 11, match the aircraft slightly better after our modifications to the activation scheme, as the PDF of the cloud droplet number concentration assembled from the model grid cells narrows slightly. However, the width of the PDF is still biased to be too wide and sometimes too concentrated at low droplet number concentrations.

Table 4 shows a comparison of the simulated and observed width of the spatial distribution of the in-cloud updraft speed before scaling. For the 500 m-resolution simulation, the factor two correction is clearly appropriate, except at 1900 m altitude

**Table 4.** Standard deviation of in-cloud updraft speed in observations and in simulations at four different resolutions, on 19 August 2017 at 1200 UTC.

| Altitude (m) | $\sigma_w$ obs. (m s$^{-1}$) | $\sigma_w$ 200 m (m s$^{-1}$) | $\sigma_w$ 500 m (m s$^{-1}$) | $\sigma_w$ 1.5 km (m s$^{-1}$) | $\sigma_w$ 3 km (m s$^{-1}$) |
|---|---|---|---|---|---|
| 800 | 1.23 | 0.69 | 0.61 | 0.57 | 0.40 |
| 1450 | 1.39 | 0.73 | 0.58 | 0.42 | 0.32 |
| 1900 | 1.52 | 0.88 | 0.44 | 0.20 | 0.17 |
| 2215 | 1.60 | 1.30 | 0.86 | 0.47 | 0.25 |
| 2550 | 2.68 | 1.62 | 1.32 | - | 1.12 |

where it is too small. By contrast the correction of a factor of three suggested by the parameterization of Malavelle et al. (2014) would be too large. We also ran simulations with 200 m, 1.5 km and 3 km resolution. For the 3 km simulation, where the Malavelle et al. (2014) scheme would suggest a scaling factor of about 8 is needed, and we find similarly that for most altitudes this would increase the width of the updraft PDF substantially beyond that which is observed (Table 4 and Supplementary Figure S13). Similarly, at 200 m resolution the factor is between 2 and 3, which is also too high. Further work is needed to understand the reasons for this overestimated correction factor fully, but the underlying premise that the updraft may be scaled up to account for its unresolved fraction may still be appropriate.

Our improved activation scheme gives a modest improvement to model performance in the cloud we chose, but we have not demonstrated it would work in any cloud. In a thin cloud with high cloud fraction but low liquid water content, the relaxation time might reach values comparable to the model timestep, which will most likely lead to biases. To avoid unphysical results, the quasi-steady state scheme could be switched off if the relaxation time is equal to, or exceeds, the timestep, and the ARG scheme used by itself instead. In our simulation, the relaxation time is either well below the timestep, or the quasi-steady-state supersaturation is higher than the ARG supersaturation, and in this case the quasi-steady-state supersaturation is not used. However, there is no guarantee that these conditions would always be satisfied with different timestep lengths or in different cloud types.

The effect on cloud microphysics of the changes to the activation scheme is detailed in Figure 12. Reducing the cloud droplet number concentration by accounting for the effect of existing clouds on supersaturation increases rain number and mass concentration, unsurprisingly, and increasing the updraft speed in the activation scheme reduces rain formation in the very highest clouds. However, the impact of the results on microphysics is small: in particular, there is no substantial impact on cloud liquid water content in these simulations, and no effect on the altitude at which rain forms.

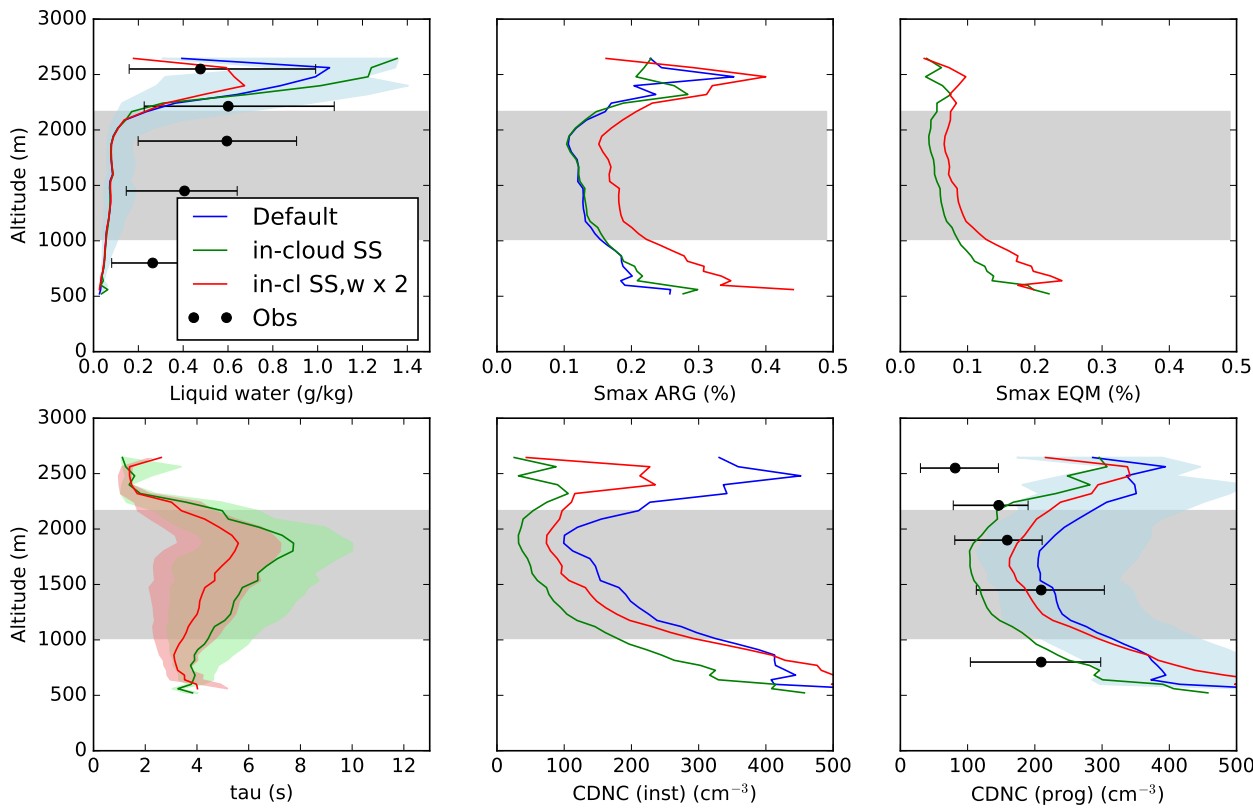

**Figure 10.** Liquid water content, maximum supersaturation 'Smax' calculated via the ARG parameterization and the in-cloud equilibrium 'EQM', supersaturation relaxation time 'tau', instantaneous concentration of new droplets activated 'CDNC (inst)' and prognostic droplet number concentration 'CDNC (prog)', at 1200 on 19 August. Each plot shows the old version of the 500 m-resolution-model, labelled 'Default', the case where the reduction of supersaturation in clouds is accounted for, 'in-cloud SS', and the cases where this reduction is accounted for and the updraft speed in the activation scheme is increased by a factor 2, 'in-cl SS, w×2'. In-cloud medians are shown (so the liquid water content is above $0.01\,\mathrm{gkg^{-1}}$ and we divide by the cloud fraction), and, except in the case of liquid water content and prognostic cloud drop number (the top-left and bottom-right) the plots also only show means over grid cells where a positive water mass is condensing. The observed median liquid water content and droplet concentrations are shown as black dots on the first and last subfigures. Where shown, error bars or shading indicate the interquartile range either of the sampled observations of the model grid cells. The plot of the relaxation time tau only includes grid cells where the relaxation time is used - i.e. where the cloud fraction exceeds 0.05 and the equilibrium supersaturation in the cloudy part of the grid cell is lower than the ARG supersaturation. We also show shaded in grey the vertical band containing 95% of the total liquid water content, because otherwise the eye is drawn to the higher altitudes. Above the shaded region, there are only a few cloudy grid cells, which do not have a strong impact on the domain-mean radiative properties, because the tops of most of the clouds in the domain are at or slightly below the top of the shaded region, not above it.

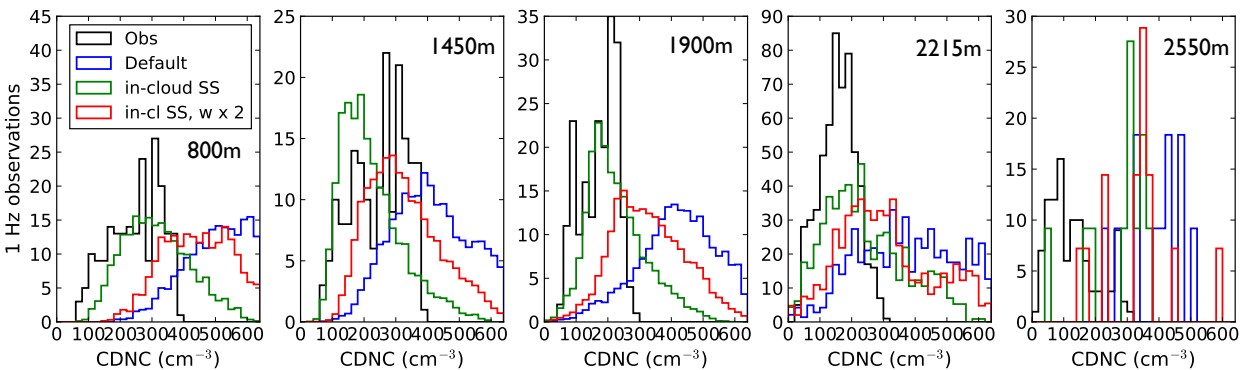

**Figure 11.** Comparison of in-cloud cloud droplet number distributions in the five straight-and-level runs at altitudes of 800, 1450, 1900, 2215 and 2550 m from left to right. Each plot shows the old version of the model, labelled 'Default', the case where the reduction of supersaturation in clouds is accounted for, 'in-cloud SS', and the cases where this reduction is accounted for and the updraft speed in the activation scheme is increased by a factor 2, 'in-cl SS, w×2'. Only model grid-boxes and CDP observations where the liquid water content exceeds $0.15\,\mathrm{g\,kg^{-1}}$ are shown.

**Table 5.** Mean in-cloud cloud droplet number concentration in the three simulations in Figures 10 and 11 (original, supersaturation corrected in clouds, and supersaturation corrected and updraft speed increased) and in the observations from the FAAM aircraft cloud droplet probe. Model grid-boxes and CDP observations where the liquid water content exceeds $0.15\,\mathrm{g\,kg^{-1}}$ are shown above the horizontal line, those with lower liquid water contents below.

| Altitude (m) | Obs. $\mathrm{cm^{-3}}$ | Default $\mathrm{cm^{-3}}$ | (in-cloud SS) $\mathrm{cm^{-3}}$ | (in-cl SS, w x 2) $\mathrm{cm^{-3}}$ |
|---|---|---|---|---|
| 800 | 249 | 512 | 316 | 445 |
| 1450 | 247 | 416 | 223 | 326 |
| 1900 | 173 | 421 | 228 | 329 |
| 2215 | 158 | 383 | 244 | 310 |
| 2550 | 126 | 388 | 277 | 329 |
| 800 | 117 | 220 | 152 | 231 |
| 1450 | 89 | 183 | 109 | 172 |
| 1900 | 45 | 174 | 98 | 147 |
| 2215 | 45 | 156 | 101 | 154 |
| 2550 | 14 | 182 | 160 | 236 |

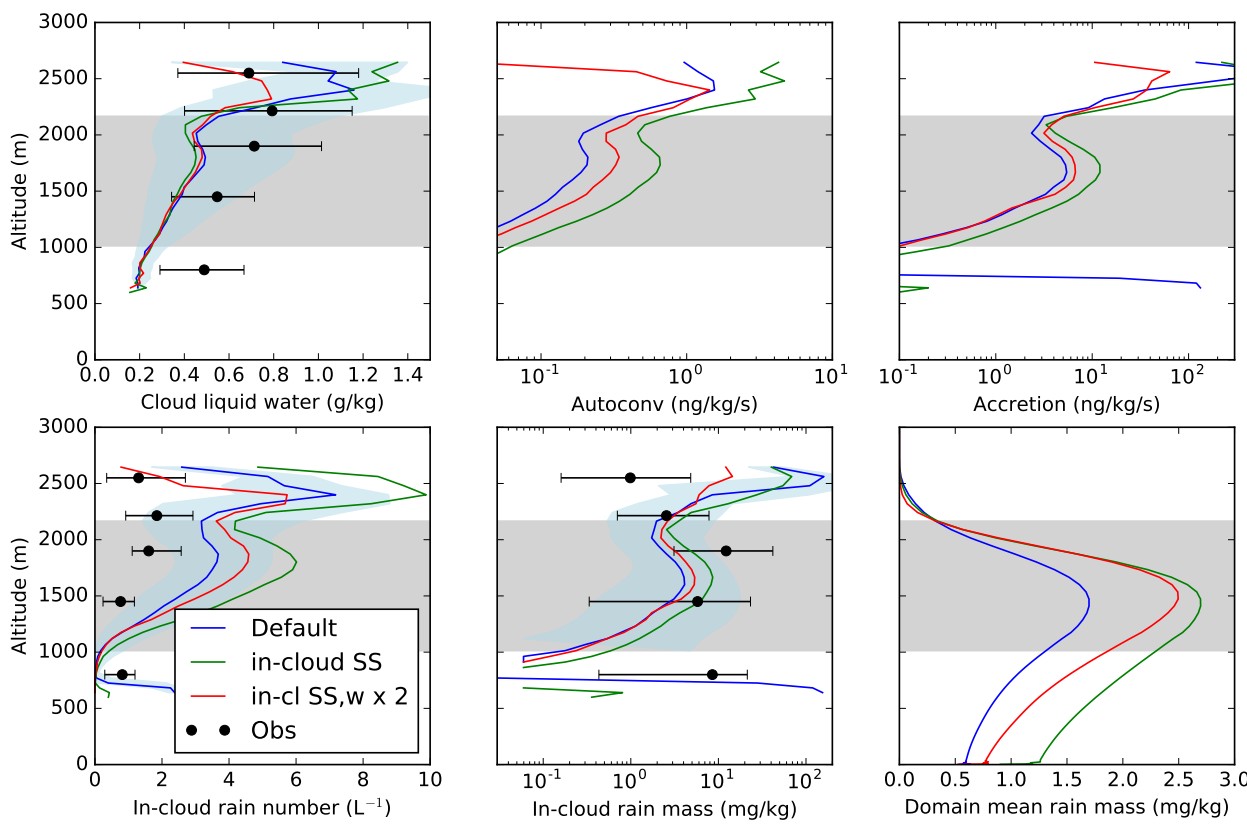

**Figure 12.** Rain and the rain formation process for observations (black dots) and model grid cells (coloured lines) where the liquid water content exceeds $0.15\,\mathrm{g\,kg^{-1}}$ in the three simulations shown in Figures 10 and 11 (original, supersaturation corrected in clouds, and supersaturation corrected and updraft speed increased). In CASIM as in many microphysics schemes, rain formation begins with an autoconversion rate, labelled 'Autoconv'. Rain droplets then accumulate liquid water by accretion. All plots except the last show in-cloud medians where we only include grid-boxes with cloud fraction greater 0.05, and the grid-box value is divided by the cloud fraction. Where shown, error bars and blue shading represents the interquartile range, while the grey shading shows the region where 95% of the domain mean liquid water content resides. The last plot shows the domain mean rain mass mixing ratio with no liquid water content threshold applied.

## 8 Discussion and conclusions

We have coupled the GLOMAP and CASIM two-moment aerosol and two-moment cloud microphysics components within the Unified Model and tested the resulting model at 500 m horizontal resolution against CLARIFY campaign data. The new configuration is intended to lead to improved simulations of aerosol-cloud interactions. We made some additional developments to the model, summarised in Table 6.

In the case study we simulated, smoke and marine aerosols are emitted and propagated to the neighbourhood of Ascension Island, where they interact with clouds and are scavenged by rain. The CLARIFY flight on 19 August 2017 has proved a useful test-bed for model evaluation. The five level in-cloud aircraft flight segments allowed large data samples to be obtained with state-of-the-art instrumentation. Additional profiles and saw-tooth segments are also available for analysis. The cloud deck, which is just under 200 km across, is substantial but self-contained, allowing plenty of data to be obtained both in and out of cloud. The case can be simulated reasonably well with our numerical weather prediction model. While other cloud cases have been measured in more detail, this is one of the best cases from the south-east Atlantic stratocumulus-to-cumulus transition zone and would be suitable for future detailed studies of aerosol-cloud interactions.

Our evaluation of aerosol and cloud microphysics is ambitious as it relies on our hierarchy of simulations accurately representing the complete cycle of aerosol emissions, transport and deposition at the synoptic scale, as well as cloud microphysical processes. We highlighted some areas where the models perform well but also some shortcomings. The 500 m-resolution model simulates the boundary layer clouds realistically, although the top of the boundary layer is low and the cloud we focus on is not quite simulated in the right place. Unfortunately, the low boundary layer top means the matching of the simulated cloud to observations at the same altitudes leads to an imperfect comparison of other cloud properties such as droplet concentration and updraft speed. However, accumulation-mode aerosol, cloud and rain number concentrations are realistic: simulated mean accumulation-mode aerosol and cloud droplet number concentrations are generally within a factor two of observations, although larger discrepancies exist in certain cloud regimes, for example, droplet number concentration is more severely overestimated in clouds with low liquid water contents. Rain mass is also simulated realistically (within a factor two) but rain number concentration is overestimated by around a factor three. Further work on the global model aerosol code is needed to address the substantial overprediction of Aitken-mode aerosol number concentrations. Cloud liquid water content and rain mass compare reasonably well to satellite measurements. We found the ARG (DIAG) aerosol activation requires a fix to work once updraft speeds are partially or fully resolved.

We examined two possible improvements to the ARG (PROG) aerosol activation scheme in CASIM: the correction to the supersaturation for existing cloud, and the Malavelle et al. (2014) correction to the updraft speeds. A gamma distribution with $\mu = 5$ was found to be needed for the first of these improvements to work, instead of the exponential distribution usually used for the CASIM cloud droplet size distribution, and so all of our results are presented with the gamma distribution. Further work is needed to ensure the Malavelle et al. (2014) correction scales with resolution correctly before it can be implemented online. Conversely, the ARG (PROG) activation scheme will break at coarser model resolution, as it is not possible to apply the Malavelle et al. (2014) correction unless some fraction of the updrafts are resolved. To get around this a sub-grid vertical

**Table 6.** Summary of model developments and adjustments documented in this manuscript

| Section | Development |
|---------|-------------|
| Section 4 | GLOMAP-CASIM coupling via activation |
| Section 4 | GLOMAP-CASIM coupling via scavenging/chemistry |
| Section 6.7 | Adjustment of $\mu$ in cloud droplet size distribution |
| Section 4 | Reduction of CASIM threshold updraft for ARG (PROG) activation |
| Section 5 | In-cloud droplet activation in ARG (PROG) activation scheme |
| Section 5 | Sub-grid updraft correction in ARG (PROG) activation scheme only |
| | implemented as a case-specific tuning factor) |

velocity of the form suggested by Morrison and Gettelman (2008), or a PDF as in ARG (DIAG), could be implemented into CASIM and switched on for low grid resolutions (probably coarser than around $3\,\mathrm{km}$, depending on the cloud type) so that aerosols will still activate when the grid-box mean updraft speed is zero. We also recommend reducing the minimum updraft speed in the ARG (PROG) activation scheme from $0.1\,\mathrm{ms}^{-1}$ to $0.001\,\mathrm{ms}^{-1}$ to avoid unphysical spikes in the distribution of
cloud droplet number.

We have now made the model more physically sound, by reducing incorrect in-cloud activation and using vertical velocities to activate aerosol that are closer to the real vertical velocities observed, and without introducing any new tuning parameters or computational expense. For high-resolution simulations, we believe the approach to secondary activation we adopted following Korolev (1995); Ming et al. (2007); Ghan et al. (2011); Yang et al. (2015) is relatively generalizable and could readily be
applied to other models that represent sub-grid cloud fraction, for example WRF-chem with Morrison and Gettelman (2008) or Gettelman and Morrison (2015) microphysics, and other activation schemes such as that of Nenes and Seinfeld (2003). We note that the Morrison and Gettelman (2008) microphysics uses an updraft speed that is already adjusted for sub-grid turbulence and so has no need of the Malavelle et al. (2014) correction. Implementing only the correction to the supersaturation, and no correction to the updrafts, would lead to lower simulated droplet concentrations.

There remain arguments for either switching off droplet nucleation above cloud base altogether, assuming the cloud droplet concentration is uniform in vertical columns above cloud base, or assuming the supersaturation is constant above cloud base as done in convective clouds in CAM5 by Wang et al. (2013). In-cloud activation leads to a broadening of the cloud droplet size distribution above cloud base, but with our approach, any new droplets activated above cloud base must follow the same size distribution as existing droplets (Khain et al., 2015). However, we have still improved on the original method where the
importance of activation above cloud base was substantially exaggerated.

The improvements to the CASIM microphysics and its ARG (PROG) activation scheme may lead to more reliable simulations of aerosol activation and aerosol-cloud interactions that depend more on the concentration of fresh aerosols at cloud base and less on the more processed aerosols found higher in the cloud than previous simulations using similar models, while still allowing activation above cloud base. However, we emphasise that we have not tested the improved model in deep or cold

clouds, and we do not suggest it could replace more sophisticated activation parameterizations, some of which already calculate in-cloud supersaturation prognostically (e.g. Fan et al., 2018). In particular, the explicit, prognostic calculation is likely to be needed for calculations of convective invigoration since it is likely that latent heat release and cloud evaporation rates will be erroneous in bulk models with saturation adjustment (e.g. Grabowski, 2007; Hill et al., 2008; Lebo et al., 2012). However,

more approximate calculations, used in many if not most convection-permitting studies, may benefit from the improvements to the procedure for aerosol activation we tested, with no additional computational cost.

## 9   Author contributions

HG, KSC, PRF and AAH developed the concepts and ideas for the direction of the paper. HG, PRF, DPG, AAH and JW coupled the GLOMAP and CASIM aerosol and cloud microphysics codes. SJA, PB, KB, IC, ZC, JT, and HW collected and

calibrated the measurement data. HG carried out and analysed the model runs, performed the model evaluation with input from PRF, SJA and PB, and wrote the paper with input and comments from KSC, PRF, SJA, PB, ZC, DPG, and AAH.

## 10   Acknowledgements

This research was funded by the Natural Environment Research Council (NERC) CLARIFY project NE/L013479/1. We thank the Facility for Airborne Atmospheric Measurements, who operate the aircraft and some of the instrumentation used for the

CLARIFY campaign, and everyone involved with the campaign. Radiosonde data was obtained from the Atmospheric Radiation Measurement (ARM) Climate Research Facility, a U.S. Department of Energy Office of Science user facility sponsored by the Office of Biological and Environmental Research. The Terra/MODIS cloud optical depth and effective radius datasets were acquired from the Level-1 and Atmosphere Archive & Distribution System (LAADS) Distributed Active Archive Center (DAAC), located in the Goddard Space Flight Center in Greenbelt, Maryland (https://ladsweb.nascom.nasa.gov/). We acknowl-

edge use of the Monsoon2 system, a collaborative facility supplied under the Joint Weather and Climate Research Programme, a strategic partnership between the UK Met Office and NERC.

## 11   Data availability

Data from the FAAM aircraft are available on the CEDA repository http://archive.ceda.ac.uk/. Simulation data are stored at the Met Office and available from the corresponding author on reasonable request. The Met Office Unified Model is available

for use under licence. A number of research organisations and national meteorological services use the UM in collaboration with the Met Office to undertake basic atmospheric process research, produce forecasts, develop the UM code, and build and evaluate Earth system models. For further information on how to apply for a licence, see

```
http://www.metoffice.gov.uk/research/modelling-systems/unified-model
```

(last access: 1 June 2020).

## 12  Competing interests

The authors declare that they have no conflict of interest.

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
