# Peer review of "Development of aerosol activation in the double-moment Unified Model and evaluation with CLARIFY measurements"

_Atmospheric Chemistry and Physics, 2020_

## Referee Comment (RC1) · Anonymous Referee #1 · 6 Apr 2020

Review of: "Improving aerosol activation in the double-moment Unified Model with CLARIFY measurements"

Authors: Gordon et al. General comment:

The authors present results for a new 2-moment microphysics scheme for activation of aerosols and nucleation of cloud droplets in the UK Met Office Unified Model. They demonstrate marginal improvements via validation of simulated marine stratocumulus clouds over the previous 1-moment scheme that provided ok results for the wrong reasons due to compensating precipitation processes. The presentation and analysis is sufficient, but I find it difficult to accept some of the justifications, simplifications,

and assumptions made in the schemes being discussed. More specific details about this statement are below. Perhaps the authors can address my comments and shed additional light on the motivation for the efforts described herein.

Specific comments:

1.Page 3, Line 27 down to Page 4, Section 2: It's unclear why you would use a 2-moment scheme with aerosols and then use a saturation adjustment for supersaturation and condensation. In an aerosol-limited regime, like typical marine stratocu, what happens if you consume all the aerosols but still have vertical motion and generation of supersaturation? If you are not permitting supersaturation to be carried around, where does the excess vapor go? To the growth of existing droplets? Will those just grow indefinitely via saturation adjustment? It seems like this could truly cause a problem with reliable prediction of latent heat release and droplet growth. I think more explanation or justification is needed beyond a reference to Morrison and Gettleman (2008).

2.Page 6, Line 33: Why do the fine grid regional domains go all the way up to 40km? Seems like you could reduce the number of vertical levels substantially and improve runtime if you topped things out in the lower stratosphere, especially since you're not simulating deep clouds.

3.Page 11, Lines 8-9: Could these high peaks be resulting from saturation adjustment? If you suddenly force new droplets to form and/or vapor growth of existing droplets in order to use up all saturation in one time-step, perhaps this is shocking the system and creating a sudden spike in drops and perhaps also a spike in latent heat release, buoyancy, and W. This is one particular reason to move away from saturation adjustment schemes.

4.Page 12, Lines 9-10: Perhaps the under-predicted RH is causing less simulated cloud fraction shown in figure 3. Maybe you're just not getting enough of an area that can generate clouds due to lack of moisture. Any thoughts?

5.Page 15, Lines 6-7: I would suspect you're not getting enough simulated convection. Your simulated cloud area is a lot less than the satellite viewed cloud cover. Perhaps this again goes back to the simulation being too dry? Your vertical velocity distributions in Figure 7 show that the simulations produce FAR fewer strong updrafts with very few instances of simulated W > 2 m/s. This would also help explain why your cloud cover is much less than observed. This could certainly be a model resolution issue and perhaps these simulations should have been run with DX < 200m in order to get better resolved updrafts.

6.Page 15, Lines 14-15: This again looks like it could be an issue with saturation adjustment in combination with activating aerosols with such a scheme.

7.Page 18, Line 3: The underestimation of large drops could be due to the required fit to a gamma distribution. It could also have something to do with rain drop breakup being too aggressive when large oblate drops attempt to form via collisions.

8.Page 21, Lines 23-24: In light of the comment regarding Phillips et al. (2007), I have to wonder why efforts are being made to improve a scheme with admittedly known limitations rather than adapting the model to something that avoids saturation adjustment and permits prognostic supersaturations. Many models have already moved in this direction.

9.Page 23, Lines 3-4: Here you state that the "ratcheting" effect should cause the 2-moment scheme to produce more droplets than the 1-moment scheme. Many times, 1-moment schemes hold the number concentration fixed, so your statement isn't really broadly true.

10.Page 25, Lines 23-25: If the PDFs are different, this is likely due to your choice of gamma distribution shape parameter. You would get a different solution if you change the breadth of the distribution.

---

## Referee Comment (RC2) · Anonymous Referee #2 · 18 Apr 2020

This study introduced the first configuration of the UK Met Office Unified Model in which both cloud and aerosol particles have 'double-moment' representations and evaluated the model at two different resolutions – 7 km and 0.5 km. The authors also made changes to aerosol activation to improve the representation of this process. The changes in calculating activation based on existing drops and in accounting for the effect of unresolved vertical velocities are trivial, in my opinion. The work is valuable for the UK Met Office Unified Model, and testing climate physics parameterizations at high-resolution (CRM) is also a meaningful try. But I did not see good model results The detailed model evaluation with aircraft measurements is also valuable. The current version needs major revisions to be accepted as a publication in ACP. Here are

the major concerns.

(1) The writing of the manuscript is sloppy. Many statements are not in the format of scientific writing. Also, there are many confusing statements and confusing use of the terms. Simulations are not clearly described. There are inconsistent and undefined names used in in figures legends, contradicted statements, lack of clear description of data sampling/processing approach to compare with the model simulation. A lot of my detailed comments below are about these problems. I do not think I captured all of the problems. In my opinion, the manuscript was in some draft mode and not ready for submission.

(2) The organization of manuscript needs improvement. For example, Sections 6 and 7 are purely the description of model developments and they are quite long. They should be moved to the Section 3 or 4 to have model developments described together.

(3) It is an overstatement that the effect of subgrid vertical velocities on activation is accounted. Basically what the authors did was to lower the threshold of grid mean updraft speeds used for activation. The statement makes people think that they connect the activation with the subgrid vertical velocity spectrum calculated from turbulence to get this done. Using a grid mean value of updraft speed, does not justify to say "subgrid vertical velocity". This has to be clarified throughout the paper. Otherwise it would be misleading.

(4) The model configuration and model simulations are not clearly described. Suggest to use a table to clearly show the configurations of major simulations. Some confusions arise from the misuse of abbreviations, for example, in some places, UKCA and CASIM are used for microphysics schemes, while they also used for different aerosol activation schemes. In the first sentence of Section 9, one of them is referred to as an aerosol scheme?

(5) I do not think 500m resolution is fine enough to simulate stratocumulus clouds explicitly. The turbulence is very difficult to simulate at this resolution because it is

partially resolved and a portion of it needs to be parameterized but model is difficult to know how much. This might be a reason for the poor simulation of PBL height. I would suggest run a test with resolution smaller than 250 m with a smaller domain to see how the simulated cloud and updraft can be improved, particularly the inversion height. Currently, the simulations that authors presented did not do well in simulating the clouds and had a huge problem with the Aitken mode of aerosols as well. This might not justify an acceptance of the paper due to these problems. If you do not want to further look at the aerosol problem, at least try to provide a good simulation of the clouds or find the major reasons leading to the large model-observation discrepancy.

Specific comments: 1. Need to describe explicitly the model configuration/resolution instead of using something like RA1, GA7.1 .

2. P1 Line 18-21, text is contradicted with the text on p9 Line 12. If you are testing the aerosol and chemistry component of the model at a higher resolution than has been attempted before, why do you need to specific a kappa value for activation? Then at P9 Line 25-27, it is said volume-weighted hygroscopicity is passed to CASIM for activation. Very confusing.

3. P8-9, it is not clear why ARG is implemented differently between the 7 km and 500 m resolutions.

4. P.9 Line 5-10, First, is ARG applicable to 500-m resolution since it was developed based on cloud parcel model with timesteps for global climate models? Figure 3 showed that cloud droplet numbers from 500 m resolution are worse compared with 7-km resolution, indicating the scheme might not work well for very high-resolution. Second, based on your description here about accounting for subgrid velocity effect on activation, you are not using any subgrid velocity. If you only use a grid mean value of updraft speed, which means you are only account for the impact of resolved updraft, not subgrid vertical velocity. This has to be clarified throughout the manuscript because the writing gave me impression that a particular method is employed to account for the

subgrid velocity effect on activation in this study until I read here.

5. P9 Line 34-35, it is confusing to say "CASIM microphysics code has the capability to simulate aerosol microphysical process" just because they simulate in-cloud removal and aerosol resuspension. Those processes are called aerosol-cloud interaction processes, not aerosol microphysical processes (instead they are cloud microphysical processes).

6. Figure 3 showed that the model did not capture the observed cloud well. At least try to figure out potential reasons for the large discrepancy between the model and observation.

7. P10 Line 4-7, I am confused. Here you said In the UKCA code, aerosol resuspension is not accounted, but I think UKCA is said as just an implementation method for ARG activation scheme. If it is a different microphysics scheme, why you need a different microphysics scheme from CASIM? Also, didn't you use coupled GLOMAP-CASIM for this study? Why "a coupled GLOMAP-CASIM double-moment model that includes aerosol microphysical and chemical processing is deferred to future work"?

8. Suggest use a table to clearly describe the configurations of aerosol and microphysical processes of the simulations for each domain.

9. Figure 4, What is ACC and AIT? They are not defined. From Figure 4, the model did not capture the inversion well. Is it the problem of resolution?

10. P12 Line 5-7, confusing. What boundary layer height are you talking about?

11. Sections 5.3 and 5.4, How did you sample the data from the simulations to compare with the aircraft observation? Need some description either in the figure captions or in the text.

12. Section 5.5, should we expect 500m resolution is fine enough to simulate stratocumulus clouds? See my major comments #4.

13. P15 Line 9-11, do you still have cloud fraction for 500-m resolution? If so, how is the cloud fraction determined and is the way to determine cloud fraction is the same between the 7 and 0.5 km resolution?

14. Figure 8, which simulation does "Model" denote? What are UKCA and UKCA 7 here? I could not find definition of these terms anywhere. In the Table 1, there are simulation names of CASIM, UKCA 500m, UKCA 7km, UKCA global. None of them are consistent with the names used in Figure 8.

15. The poor simulations of cloud droplet distribution might indicate that ARG scheme might not be applicable to those resolutions, as I commented above.

16. Figure 9, which simulation did you compare here? Such information should be clearly described in the figure caption.

17. P20, 11-14, which activation scheme are you talking about here? There are two activation schemes - UKCA and CASIM as discussed above.

18. The organization of Section 6-8 is strange since Sections 6 and 7 are purely the description of model developments. They should be moved to the Section 3 or 4 to have model developments described together.

19. Section 9, in-cloud activation is thought as secondary nucleation, which should increase droplet number. Here it decreases CDNC. I think here it means different from what people usually think. It is just a way to treat activation with accounting for existing droplets? Please clarify. Is it added to the primary nucleation (cloud-base nucleation)? If so, how can it reduce CDNC?

20. Section9, first sentence: it is a confusing sentence. Which one is two-moment aerosol scheme and which is the two-moment cloud microphysics? Based on what I read, UKCA and CASIM were also used for naming different activation schemes.

21. P26, Line 15-16, I do not think it is scientific writing by saying "model emits smoke..."

22. P28, Line 1, First I do not agree that the model performs well based in the results shown. Second, even if it indeed performs well, please state perform well in what quantity and what aspect. It is not a scientific writing to only state "the model performs well".

23. P28, 7, I have no idea what you mean about "cloud deck is just under 200km across".

---

## Author Response (AR1)

**Responses to Reviewer 1:**

The authors present results for a new 2-moment microphysics scheme for activation of aerosols and nucleation of cloud droplets in the UK Met Office Unified Model. They demonstrate marginal improvements via validation of simulated marine stratocumulus clouds over the previous 1-moment scheme that provided ok results for the wrong reasons due to compensating precipitation processes. The presentation and analysis is sufficient, but I find it difficult to accept some of the justifications, simplifications, and assumptions made in the schemes being discussed. More specific details about this statement are below. Perhaps the authors can address my comments and shed additional light on the motivation for the efforts described herein.

We thank the reviewer for their helpful comments. We have identified improvements we can make to our article in response to these comments and those of the other reviewer. We will highlight the main changes in red in a new version we are ready to upload.

Page 3, Line 27 down to Page 4, Section 2: It's unclear why you would use a 2-moment scheme with aerosols and then use a saturation adjustment for supersaturation and condensation.

We appreciate that some people have assumed in the past that double-moment microphysics schemes include prognostic supersaturation (we found an example in the review by Guichard and Couvreux (2017)). We therefore agree with the reviewer that we should justify our choice more explicitly, and so we propose to include the following paragraph in our new draft of the manuscript:

"In our CASIM microphysics scheme, saturation adjustment is applied. It has sometimes been assumed that prognostic supersaturation is generally part of double-moment microphysics schemes (Guichard *et al.*, 2017). However, Shipway and Hill (2012) compared several single and double-moment bulk microphysics schemes including CASIM with a bin microphysics scheme in a single-column framework. The bin scheme treated supersaturation prognostically while most of the bulk schemes did not. The double-moment bulk schemes with saturation adjustment they tested were in closer agreement with the bin microphysics scheme than the single-moment schemes. Their conclusions were substantiated further by Hill *et al* (2015). While useful, prognostic supersaturation is not essential for a double-moment microphysics scheme to improve on a single-moment scheme."

We understand that saturation adjustment can still lead to biases, which we discuss below. A version of the UM with prognostic supersaturation would be a valuable research tool that would allow us to reach shorter timesteps and higher spatial resolutions, and we would like to implement it in future. However, we are not trying to simulate convective invigoration by aerosols with high accuracy (which is where we feel prognostic supersaturation is most important in representing aerosol-cloud interactions in models), as we pointed out in the text.

There is clear value in a one-size-fits-most weather- and climate model microphysics parameterization that can work at degree-scale resolution down through almost all the numerical weather prediction scales to about 500m resolution. The configuration of the model we introduce in this paper will allow improved representations of (for example) precipitation, fog, or the Twomey effect in weather and climate simulations, almost all of which are run at lower spatial resolution than we use here. We are testing the model at the finest resolution we think the

microphysics with saturation adjustment should be able to handle for these clouds. We emphasise, though, that we present a new configuration of an existing numerical weather prediction (NWP) model, and the GLOMAP aerosol and CASIM cloud microphysics schemes have been characterized before.

If we used prognostic supersaturation, we also would need a timestep of order 2s which would be unfeasible in our climate and NWP models. In the model of Morrison and Grabowski (2008) with prognostic supersaturation, timesteps longer than 3 seconds were shown to lead to substantial biases at realistic updraft speeds. Lebo and Morrison (2014) similarly use 3 seconds. Other schemes, for example that of Phillips et al (2007), use microphysical substeps, and while the scheme in the RAMS model was demonstrated to work with 10s timesteps in a simple case (Walko *et al.*, 2000) even 10s is much too short for an NWP model.

In the abstract, we said previously that our improvements
'reinforce our confidence in the ability of the model to simulate aerosol-cloud microphysical interactions'. To avoid giving the impression that the model is capable of representing *all* aerosol-cloud microphysical interactions, we plan to rephrase this to say 'simulate the aerosol-cloud microphysical interactions it was designed to represent'.

In an aerosol-limited regime, like typical marine stratocu, what happens if you consume all the aerosols but still have vertical motion and generation of supersaturation? If you are not permitting supersaturation to be carried around, where does the excess vapor go? To the growth of existing droplets?
Yes, excess vapor would still lead to droplet growth. The only issue with saturation adjustment in itself is that this condensation would occur too quickly, and therefore at the wrong altitude, if the timesteps are short. However, we show that for our model the timesteps are (almost always) not too short: the relaxation time (given in our paper, at page 20, line 33 of the posted discussion article) for supersaturation is almost always shorter than the timestep (Figure S13, top-left). However, we agree that the potential biases would still be worth investigating in future work.
Will those just grow in-definitely via saturation adjustment?
No, because the water vapor will be depleted. To a reasonable approximation, for the 20s timesteps used in this model, the same amount of water condenses whether one uses saturation adjustment or prognostic supersaturation.

It seems like this could truly cause a problem with reliable prediction of latent heat release and droplet growth. I think more explanation or justification is needed beyond a reference to Morrison and Gettleman (2008).

We agree, and we propose to point out that the relaxation time constraint is satisfied in our introduction, and explain the significance of this. We also propose to add a citation to the review of cloud-resolving model microphysics schemes by Khain *et al* (2015) to back up our assertion that saturation adjustment is commonly applied, even at cloud-resolving resolution.

If supersaturations in our case study were higher, as in deep convective clouds, the relaxation time would be longer, and we would be more likely to incur biases if we did not lengthen the timestep and reduce the spatial resolution, possibly to the point that the convection would not be sufficiently well resolved. Lebo et al (2012) suggest one minute is a sufficiently long timestep for saturation adjustment to work in deep convective clouds. We have not shown our model is

suitable for simulations of deep convection at our 500m spatial resolution, and we pointed this out in our conclusions, at line 30 on page 21 of the posted discussion article.

It is important to note that in this work, as in many operational systems, a sub-grid cloud scheme is employed to represent sub-grid inhomogeneities in RH and cloud (even at 500m resolution). This means that cloud can exist under sub-saturated conditions. At the end of each timestep, we therefore intend 'saturation adjustment' to mean that the grid-cell mean relative humidity is not constrained to be 100%, but must instead be a grid-mean relative humidity below 100%. We appreciate that this wasn't as clear as it should have been in the discussion paper, so we plan to clarify it during revision. We will also avoid using the term 'cloud-resolving', as this could misleadingly suggest there is no sub-grid cloud scheme. The minimum relative humidities are fixed parameters in the model which may vary with altitude.

In this way, the sub-grid cloud scheme allows us to account, to some extent, for unresolved vertical velocities when we handle condensation of water vapor. Most models with prognostic supersaturation we are aware of don't represent sub-grid fractional cloud cover or the variability of supersaturation within a grid cell– there is, as far as we know, only one prognostic variable for supersaturation, and (for example) no second prognostic for the standard deviation of its spatial distribution. Therefore, at kilometer-scale resolution, the grid-mean supersaturation these models calculate may not be representative of the real supersaturation within different parts of the grid cell, as the updrafts from which the prognostic supersaturation is calculated are not resolved. This potential source of bias may be large – as model grid cell size is increased, at some point it would become larger than biases from saturation adjustment – and it is avoided by using saturation adjustment with a sub-grid cloud scheme.

Page 6, Line 33: Why do the fine grid regional domains go all the way up to 40km? Seems like you could reduce the number of vertical levels substantially and improve runtime if you topped things out in the lower stratosphere, especially since you're not simulating deep clouds.

We agree the number of vertical levels is excessive for this case. The high model top is the default in regional UK Met Office operational NWP configurations such as UKV or RAL (Bush *et al.*, 2020) and we did not need to change it for this study.

Page 11, Lines 8-9: Could these high peaks be resulting from saturation adjustment? If you suddenly force new droplets to form and/or vapor growth of existing droplets in order to use up all saturation in one time-step, perhaps this is shocking the system and creating a sudden spike in drops and perhaps also a spike in latent heat release, buoyancy, and W. This is one particular reason to move away from saturation adjustment schemes.

We think this is possible, as occasionally the grid-mean relaxation time exceeds the model timestep. However, this is a rare occurrence. We propose to add a sentence "It is possible that these peaks correspond to the few grid cells where our saturation adjustment assumption has led to excessive release of latent heat, temporarily strengthening the updrafts."

Page 12, Lines 9-10: Perhaps the under-predicted RH is causing less simulated cloud fraction shown in figure 3. Maybe you're just not getting enough of an area that can generate clouds due to lack of moisture. Any thoughts?

This is possible, though we cannot be sure the isolated measurement in a single column of the atmosphere is representative. See our next comment.

Page 15, Lines 6-7: I would suspect you're not getting enough simulated convection. Your simulated cloud area is a lot less than the satellite viewed cloud cover. Perhaps this again goes back to the simulation being too dry?

This is a possibility. The small domain size means there is a substantial effect from boundary conditions too. We re-ran a simulation at 500m resolution with the values of the critical relative humidity in our sub-grid cloud scheme reduced by 10%, to a minimum of 80% instead of 90%. With this crude tuning we were able to produce more cloud, as shown in the center two plots of the figure below (the left two plots are the default as in Figure 3 of our paper, the center two are with the reduced critical relative humidity). We prefer to keep the standard model configuration for the main results in our paper, however, as this tuning is not likely to be relevant in other case studies and we feel it is more useful to demonstrate the performance of a well-documented model configuration. However, we will also add the figure to the supplementary materials and refer to it in the paper text.

[Figure]

*Figure 1: Comparison of the 500m-resolution model cloud liquid water path (LWP) and cloud droplet number concentration (CDNC) to MODIS AQUA observations (right) on 19 August 2017 at 1200. The leftmost plots show the default model and the center plots show the model when the critical relative humidity in the sub-grid cloud scheme is reduced by 0.1. Isolated MODIS pixels are enlarged during regridding as described in the manuscript text, and white space in the plot of cloud droplet number concentration may indicate both cloud-free areas and failed retrievals.*

Your vertical velocity distributions in Figure 7 show that the simulations produce FAR fewer strong updrafts with very few instances of simulated W > 2 m/s. This would also help explain why your cloud cover is much less than observed. This could certainly be a model resolution

issue and perhaps these simulations should have been run with DX < 200m in order to get better resolved updrafts.

We have a sub-grid cloud scheme, so in principle we shouldn't need to fully resolve vertical velocities in order to produce clouds.

Page 15, Lines 14-15: This again looks like it could be an issue with saturation adjustment in combination with activating aerosols with such a scheme.

We believe saturation adjustment would be unlikely to lead to such biases in our case. The existing text of the paper does acknowledge the inaccuracies associated with saturation adjustment in the introduction, and we discuss shortcomings in the activation scheme in Section 6 in the discussion article.

Page 18, Line 3: The underestimation of large drops could be due to the required fit to a gamma distribution. It could also have something to do with rain drop breakup being too aggressive when large oblate drops attempt to form via collisions

Yes, we agree – thank you. We will add in the point about the gamma distribution.

8.Page 21, Lines 23-24: In light of the comment regarding Phillips et al. (2007), I have to wonder why efforts are being made to improve a scheme with admittedly known limitations rather than adapting the model to something that avoids saturation adjustment and permits prognostic supersaturations. Many models have already moved in this direction.

We are using an operational NWP and climate model suitable for global to regional scales, and we test here it at a horizontal resolution that is close to the highest spatial resolution the model is run at operationally. In such a model, saturation adjustment is an appropriate approximation. Our aim is not to build a model specifically for kilometre-scale simulations of clouds.

9.Page 23, Lines 3-4: Here you state that the "ratcheting" effect should cause the 2-moment scheme to produce more droplets than the 1-moment scheme. Many times,1-moment schemes hold the number concentration fixed, so your statement isn't really broadly true

This is a good point that we overlooked. We plan to adjust the text to read:
"However, the ratcheting means the double-moment scheme should produce more droplets on average than a single-moment scheme, provided the number concentration of droplets is diagnosed from the aerosol concentration in the single-moment scheme. For example, if the diagnostic UKCA activation scheme were fed by the same updraft speeds and the same aerosol as a double-moment scheme, fewer droplets on average would be produced."

10. Page 25, Lines 23-25: If the PDFs are different, this is likely due to your choice of gamma distribution shape parameter. You would get a different solution if you change the breadth of the distribution.

We meant the PDFs of how CDNC varies in space, rather than the droplet size distribution function. Sorry for the confusion, we will amend the text to make this clear.

**Responses to Reviewer 2**

The writing of the manuscript is sloppy. Many statements are not in the format of scientific writing. Also, there are many confusing statements and confusing use of the terms. Simulations are not clearly described. There are inconsistent and undefined names used in in figures legends,

contradicted statements, lack of clear description of data sampling/processing approach to compare with the model simulation. A lot of my detailed comments below are about these problems. I do not think I captured all of the problems. In my opinion, the manuscript was in some draft mode and not ready for submission.

We thank the reviewer for taking the time to make the detailed suggestions below, which we have addressed. In addition, we have tried to make other things in the manuscript text clearer following the examples highlighted by the reviewer. We will highlight the main changes in red in a new uploaded version of the manuscript.

The organization of manuscript needs improvement. For example, Sections 6 and 7are purely the description of model developments and they are quite long. They should be moved to the Section 3 or 4 to have model developments described together.

We agree, and plan to move these sections, distributing some of the text elsewhere in the manuscript where appropriate. We also substantially restructured the description of the model in our latest draft.

It is an overstatement that the effect of subgrid vertical velocities on activation is accounted. Basically what the authors did was to lower the threshold of grid mean updraft speeds used for activation. The statement makes people think that they connect the activation with the subgrid vertical velocity spectrum calculated from turbulence to get this done. Using a grid mean value of updraft speed, does not justify to say "subgrid vertical velocity". This has to be clarified throughout the paper. Otherwise it would be misleading.

As well as lowering the threshold of grid mean updraft speeds, we also examined the procedure of Malavelle *et al* (2014) for correcting updraft speeds to account for unresolved turbulence, and while we didn't actually implement the procedure, we did derive a reasonable ad-hoc tuning factor. We agree that the ad-hoc nature of our correction, which would only work for our specific case, may not have been as clear as it should have been, and we will adjust the text appropriately.

(4) The model configuration and model simulations are not clearly described. Suggest to use a table to clearly show the configurations of major simulations. Some confusions arise from the misuse of abbreviations, for example, in some places, UKCA and CASIM are used for microphysics schemes, while they also used for different aerosol activation schemes. In the first sentence of Section 9, one of them is referred to as an aerosol scheme?

We agree our previous labels were confusing. We propose to make numerous improvements and include a table at the start of section 5, including the simulations at their three resolutions, and revise the acronyms. We occasionally used UKCA and GLOMAP interchangeably, which could cause confusion, and we also used 'UKCA activation scheme' to refer to the diagnostic implementation of Abdul-Razzak & Ghan (2000) by West *et al* (2014) and CASIM activation scheme to refer to the prognostic implementation. We now use ARG (DIAG) in place of 'UKCA activation' and ARG (PROG) in place of 'CASIM activation' throughout the paper to avoid this confusion.

(5) I do not think 500m resolution is fine enough to simulate stratocumulus clouds explicitly. The turbulence is very difficult to simulate at this resolution because it is partially resolved and a portion of it needs to be parameterized but model is difficult to know how much. This might be a reason for the poor simulation of PBL height. I would suggest run a test with resolution smaller than 250 m with a smaller domain to see how the simulated cloud and updraft can be improved, particularly the inversion height.

As we use a sub-grid cloud scheme, we are not technically trying to simulate the clouds explicitly. However, we did use the term 'cloud-resolving' in the text, which we appreciate may mislead the reader to suggest that the clouds *are* simulated explicitly. We will replace it.

We were able to make modest improvements to the simulation of the clouds by tuning the critical relative humidity in the sub-grid cloud scheme, please see our response to reviewer 1.

While the global and 500m simulations have higher PBL heights than the 7km resolution model, the variability between the three simulations is relatively small. It takes about 4 hours for air masses to advect from the edge of the 500m-resolution model domain to the center, assuming an 8m/s horizontal wind speed. As this is not very long, we think the biases in the PBL height in the 500m model are largely due to the biases in the lower-resolution driving simulations. Therefore running a higher-resolution inner simulation is unlikely to produce a substantial improvement.

The biased inversion height is partially an issue with the reanalysis that our global model is nudged to, as well as the PBL scheme. Our nudging is not too constraining: we nudge to horizontal wind, not temperature, above approximately 2000m altitude, but we note the PBL height is higher than 2000m here. We added ERA-interim and ERA5 reanalysis temperature profiles to our existing temperature profile figure (shown as Figure 2 in these replies, below), to demonstrate the bias in the reanalysis. We then plan to add the global model to Figure 4 in the main text of the paper, to show that it, too is biased, in line with the reanalysis. We will make appropriate comments in the manuscript text. We still don't know why the 7km-resolution model produces a slightly lower PBL height than the global model.

[Figure]

*Figure 2: A copy of the temperature profiles (left) shown in the manuscript with ERA5 and ERA-interim reanalysis temperature profiles superposed, together with the simulated relative humidity*

*in the three models (right). Dotted lines indicate one standard deviation across the model grid cells sampled.*

After several unsuccessful attempts to improve the 7km and global model performance, we tried running the global model nudged only above 4000m altitude (starting from 11 August), with a tuning of entrainment parameters in the boundary layer scheme, and we do see a modest 100m improvement in the boundary layer height compared to the reanalysis (Figure 3 in these replies, below). We conclude that the global model *can* be tuned to perform marginally better than either ERA-interim or ERA5 in this case, although it still underestimates the boundary layer height. The improvement in performance in this region may well be accompanied by a degradation elsewhere – we didn't check. As in the case of our tuning of the critical relative humidity (response to reviewer 1), we prefer to retain our default simulations in the paper, to avoid hiding the shortcomings of the well-defined CMIP6 or RA1 climate and weather prediction models by tuning.

[Figure]

*Figure 3: Temperature and relative humidity profiles in the tuned global model (purple), nudged only above 4000m altitude, compared to FAAM aircraft observations and, for temperature, ERA-5 and ERA-interim reanalysis in the domain of our 500m-resolution model. Dotted lines indicate one standard deviation across the model grid cells sampled.*

Currently, the simulations that authors presented did not do well in simulating the clouds and had a huge problem with the Aitken mode of aerosols as well. This might not justify an acceptance of the paper due to these problems. If you do not want to further look at the aerosol problem, at least try to provide a good simulation of the clouds or find the major reasons leading to the large model-observation discrepancy

The overestimated Aitken mode number is a problem with the global climate model, as shown by Mulcahy et al (2020), page 33 and 34 (the GC3.1 climate model has identical aerosol microphysics to our model configuration). We are preparing a global modeling manuscript

(Ranjithkumar, Gordon, Carslaw *et al*, in prep.) to study the issue more closely with the UKESM configuration, which has coupled online oxidation chemistry but is otherwise similar. We believe the overestimated concentration is the result of excessive new particle formation in the upper troposphere. We show the concentration of Aitken-mode particles in the global model at high altitudes in Figure 4 below, which we propose to include in supplementary text.

In total, including the nucleation mode (all particles above 3nm diameter), we simulate about 20000 particles per cubic centimeter at STP in the upper troposphere at 10km altitude, which is about a factor of four higher than measurements from the ATom campaign published by Williamson *et al* (2019) suggest. We note our Figure 4 shows ambient concentrations in the Aitken mode only. At 10km altitude, our $SO_2$ concentrations are about 40ppt, which is about a factor of two too high, presumably due to underestimated scavenging of $SO_2$ in deep convective clouds, erroneous emissions or deposition, or in-cloud aqueous reactions. Since new particle formation is non-linear in gas concentrations, we think the $SO_2$ may well explain the discrepancy in Aitken-mode aerosol number concentration.

[Figure]

*Figure 4: vertical profiles of aerosol number concentration in the Aitken (AIT) and accumulation (ACC) modes of the regional models with 500m and 7km resolution, and the global model (GLM), compared to the measurements from the aircraft CPC and PCASP instruments. Compared to the manuscript, we also plot simulated number concentrations from the global model at higher altitudes, showing the excessive new particle formation in the upper troposphere.*

At our simulated cloud top, about 2100m altitude, our Figure 4 shows the total Aitken mode concentration exceeds the accumulation mode concentration. However, Figure 3 in our manuscript shows that the simulated Aitken mode diameter is about 30nm at 1900m altitude, and 40nm at 2550m, so only a fraction of the Aitken mode will be large enough to activate. Moreover, most cloud droplets are not formed at the cloud top height, rather nearer cloud base, where the concentration in the Aitken mode is negligible. We think the biased Aitken mode may be partly responsible for the overestimated droplet concentration, and the overestimated width in the spatial distribution of droplets, at 2215m altitude, shown in Figure 8, and we will add a comment to that effect in the text. We will also make some relevant comments in the text that was in Section 5.3 of the discussion article to accompany the new supplementary figure.

1. Need to describe explicitly the model configuration/resolution instead of using something like RA1, GA7.1

We agree, and will restructure the description to make it clearer, add additional references to our emissions datasets, the requested summary table, and some additional detail. We prefer to keep the references to standard configurations to avoid lengthening the manuscript excessively.

2. P1 Line 18-21, text is contradicted with the text on p9 Line 12. If you are testing the aerosol and chemistry component of the model at a higher resolution than has been attempted before, why do you need to specific a kappa value for activation? Then at P9Line 25-27, it is said volume-weighted hygroscopicity is passed to CASIM for activation. Very confusing.

We use the term 'volume weighted hygoscopicity' when we should have said the 'volume-weighted kappa value' – since each component of aerosols (e..g sulphate, organic carbon) has its own assigned kappa value, and the overall kappa value of the aerosol is the volume-weighted average of the kappa values of the components. Sorry, this will be corrected. However, we still need to specify a kappa value, because we still use the activation parameterization of Abdul-Razzak and Ghan (2000).

P8-9, it is not clear why ARG is implemented differently between the 7 km and 500m resolutions.

The reason is that the 7km and the global simulations still use single-moment cloud microphysics, while the 500m-simulation uses double-moment cloud microphysics (the CASIM scheme). This will be clarified in the new text.

4. P.9 Line 5-10, First, is ARG applicable to 500-m resolution since it was developed based on cloud parcel model with timesteps for global climate models?

We did not identify any physical reason why ARG should not be run in models with higher horizontal resolution (and the vertical resolution in our model is also closer to that of a climate model than to a large eddy simulation.) We don't believe the reviewer meant that the cloud parcel model itself could have used timesteps from global models (~20 minutes). In fact, we would expect ARG to perform better at higher spatial resolution, because a higher fraction of the sub-grid variability in aerosol concentrations which, unlike the variability in updraft speeds, are not usually accounted for, will be resolved.

The ARG parameterization or similar parameterizations have been applied successfully in cloud-resolving models, or models with similar spatial resolution, before, as summarised in Table 3 of Ghan *et al* (2011). We will note this in the text.

Figure 3 showed that cloud droplet numbers from 500 m resolution are worse compared with 7-km resolution, indicating the scheme might not work well for very high-resolution.

It's true that the cloud droplet numbers are not in good agreement with observations, but we do think the underlying activation parameterization should work fine at this resolution, as discussed above. We think the biases in the cloud droplet concentration are due at least in part to biases in the accumulation-mode aerosol concentrations. In Figure 3, the 500m simulations use CASIM double-moment microphysics, and the cloud droplet numbers come from the "CASIM activation scheme" (which we propose to relabel ARG(PROG)) while those in the center from the 7km simulations use Wilson and Ballard single-moment microphysics and the "UKCA activation scheme" that we propose to relabel ARG(DIAG), so unfortunately the activation schemes cannot be directly compared in this figure.

Second, based on your description here about accounting for subgrid velocity effect on activation, you are not using any subgrid velocity. If you only use a grid mean value of updraft speed, which means you are only account for the impact of resolved updraft, not subgrid vertical velocity. This has to be clarified throughout the manuscript because the writing gave me impression that a particular method is employed to account for the subgrid velocity effect on activation in this study until I read here.

In our default simulation, we do not use a subgrid velocity. We will clarify this in the manuscript. We do, however, show results from a simulation where we scale the grid-box mean vertical velocity to illustrate what a simulation that accounted for sub-grid effects might look like. Malavelle *et al* (2014) derive a scaling factor for the variability in vertical velocity with the appropriate dependence on model grid resolution. We assert in this paper that, if we average over many model timesteps and spatial grid cells, scaling the grid-mean updraft velocity should have the same effect as scaling the variability in vertical velocity, provided, in line with Malavelle *et al*, that enough of the variability in updraft speed is resolved by the grid. For no bias to result, the vertical velocity would need to vary independently amongst the model grid cells, and from one time-step to the next, and it would need to have a Gaussian distribution when many grid cells and timesteps were combined. This is not a very good approximation, but the simulation is still a reasonable demonstration of the effect expected from a sub-grid velocity distribution.

5. P9 Line 34-35, it is confusing to say "CASIM microphysics code has the capability to simulate aerosol microphysical process" just because they simulate in-cloud removal and aerosol resuspension. Those processes are called aerosol-cloud interaction processes, not aerosol microphysical processes (instead they are cloud microphysical processes)

We agree that our use of "processing" here is confusing, as not all readers may be familiar with the jargon. We will change the text to read 'cloud microphysical processing of aerosol'– thank you.

Figure 3 showed that the model did not capture the observed cloud well. At least try to figure out potential reasons for the large discrepancy between the model and observation

Please see our response to reviewer #1 and major comment #5.

P10 Line 4-7, I am confused. Here you said In the UKCA code, aerosol resuspensionis not accounted, but I think UKCA is said as just an implementation method for ARG activation scheme. If it is a different microphysics scheme, why you need a different microphysics scheme from CASIM? Also, didn't you use coupled GLOMAP-CASIM for this study? Why "a coupled GLOMAP-CASIM double-moment model that includes aerosol microphysical and chemical processing is deferred to future work

We will change the acronyms, as outlined in our response to major comment #4.

8. Suggest use a table to clearly describe the configurations of aerosol and microphysical processes of the simulations for each domain

This is a good suggestion, we will add the table.

9. Figure 4, What is ACC and AIT? They are not defined. From Figure 4, the model did not capture the inversion well. Is it the problem of resolution?

We will define the abbreviations in the caption. For the inversion, please see above.

10. P12 Line 5-7, confusing. What boundary layer height are you talking about?

Ascension Island is not in our simulation domain. During the case study period, the boundary layer height (BLH) at Ascension Island is 500m lower than it is in the domain of the 500m-resolution model. In the 7km resolution model, the simulated BLH at Ascension Island is 300m lower than it is in the domain of the 500m-resolution model. We agree it was written confusingly - we will clarify the text.

11. Sections 5.3 and 5.4, How did you sample the data from the simulations to compare with the aircraft observation? Need some description either in the figure captions or in the text.

For Figure 4, we used a horizontal mean of the 500m-model domain at midday UTC on 19 August 2017, after removing all grid cells within 10km of the domain boundaries.

We assume a) that the duration of the flight is sufficiently short that the observations may be compared to this temporal snapshot of the model. The aircraft entered the simulation domain at 1103 UTC and left it at 1310 UTC. We will point this out in the text, in a new introduction to the Section 5 of the discussion article. We also assume b) that variability in aerosol concentrations, temperature and relative humidity in the simulation output across the domain is quite small. The accumulation mode number varies from 600/cc to 800/cc in the BL (Figure 2) so this does introduce an error. However, because the simulated and observed clouds are not in the same place in the domain, we think it is better to use a spatial average than to sample the simulation along the track of the aircraft. In the figure caption, we plan to say: "Variability across the model domain is assumed here to be small, but, for accumulation-mode aerosol, it may be inferred from Figure 2."

Figure 5: we will point out the simulated data are "a mean over the domain at 1200 UTC. Only aircraft data from straight-and-level legs at the specified altitude are shown."

Figure 6: in the caption we will additionally specify: "We include all model grid cells at the specified altitude within the simulation domain with a liquid water content greater than $0.01gkg^{-1}$ at the instant of 1200 UTC on 19 August in the histogram, except for the grid cells within 10km of the domain boundary."

Figure 7: We will add "Model data are from instantaneous simulation output of the 500m and the 7km resolution simulations, in both cases across the domain of the 500m simulation, at 1200 UTC on 19 August 2017."

12. Section 5.5, should we expect 500m resolution is fine enough to simulate stratocumulus clouds? See my major comments #4

Please see response to major comment #5.

13. P15 Line 9-11, do you still have cloud fraction for 500-m resolution? If so, how is the cloud fraction determined and is the way to determine cloud fraction is the same between the 7 and 0.5 km resolution?

We do have cloud fraction, determined diagnostically using the scheme of Smith (1990), and it is the same at both model resolutions. This is now included in the model description. We now avoid using the term 'cloud-resolving' as this may have implied we did not have a cloud fraction.

14. Figure 8, which simulation does "Model" denote? What are UKCA and UKCA 7here? I could not find definition of these terms anywhere. In the Table 1, there are simulation names of CASIM, UKCA 500m, UKCA 7km, UKCA global. None of them are consistent with the names used in Figure 8.

We changed the labels throughout to ARG(PROG), ARG (DIAG) etc, including on the figure caption, to reflect the prognostic double-moment cloud droplet number in the activation scheme used with CASIM microphysics, or the diagnostic activation scheme used with the single-moment microphysics of Wilson and Ballard (1999). The orange and red curves in Figure 8 are

both from the 500m simulation – orange is the diagnostic activation scheme we previously called 'UKCA activation' and now call ARG(DIAG), while red is the activation scheme for cloud droplets in the double-moment CASIM microphysics scheme, now called ARG(PROG). The green curve is the UKCA (now ARG(DIAG)) activation scheme at 7km resolution.

15. The poor simulations of cloud droplet distribution might indicate that ARG scheme might not be applicable to those resolutions, as I commented above.
On the applicability, please see our response to minor comment #4. Our paper focuses on the ARG(PROG) scheme, but the ARG(DIAG) scheme also has shortcomings, some of which are documented in the supplement. So far, however, we have only identified shortcomings in the implementation of the scheme, and not in the underlying ARG algorithm. These shortcomings are combined with biases in the simulated aerosol concentration.

16. Figure 9, which simulation did you compare here? Such information should be clearly described in the figure caption.
We agree, and propose to add 'The simulations are from the 500m-resolution model (the only simulation with double-moment cloud microphysics).'

17. P20, 11-14, which activation scheme are you talking about here? There are two activation schemes - UKCA and CASIM as discussed above
CASIM (now labelled ARG (PROG)); this will be clarified.

18. The organization of Section 6-8 is strange since Sections 6 and 7 are purely the description of model developments. They should be moved to the Section 3 or 4 to have model developments described together.
This will be done.

19. Section 9, in-cloud activation is thought as secondary nucleation, which should increase droplet number. Here it decreases CDNC. I think here it means different from what people usually think. It is just a way to treat activation with accounting for existing droplets? Please clarify. Is it added to the primary nucleation (cloud-base nucleation)? If so, how can it reduce CDNC?
In our original model, as in simulations published by Grosvenor et al (2017), Miltenberger et al (2018), and various others, "primary nucleation" was applied irrespective of whether or not a cloud is present already in the grid box. We understand this is also done in other models, such as WRF with Morrison & Gettelman (2008) microphysics. When we apply "secondary nucleation", which is indeed a way to account for existing droplets, we apply it instead of primary nucleation, and therefore the overall droplet number goes down because the supersaturation used in the activation scheme is lowered by the existing droplets.

We describe how we combine 'primary' and 'secondary' nucleation in partially cloudy grid boxes as defined by our cloud fraction scheme. It is the way of doing this combination that we feel makes our change non-trivial. We will modify some of the introductory text to make our changes clearer.

20. Section 9, first sentence: it is a confusing sentence. Which one is two-moment aerosol scheme and which is the two-moment cloud microphysics? Based on what I read, UKCA and CASIM were also used for naming different activation schemes.

We agree, and will relabel the activation schemes.

21. P26, Line 15-16, I do not think it is scientific writing by saying "model emits smoke..
We will adjust the text to 'In our simulations, smoke and marine aerosols are emitted…'

22. P28, Line 1, First I do not agree that the model performs well based in the results shown.
Second, even if it indeed performs well, please state perform well in what quantity and what
aspect. It is not a scientific writing to only state "the model performs well".
We agree, and propose to remove this comment and the subsequent text, and integrate the
content later with a more quantitative statement, as the model performance is discussed more
specifically in the paragraph that follows, starting 'Our evaluation of meteorology and of aerosol
and cloud microphysics highlighted some areas where the model performs well but also some
shortcomings.'

23. P28, 7, I have no idea what you mean about "cloud deck is just under 200km across"
The approximate spatial extent of continuous cloud cover in the horizontal direction is just under
200km. We did not understand the source of confusion here, but we are happy to amend the text
if needed.

**References for review replies**

Abdul-Razzak, H. & Ghan, S. J. A parameterization of aerosol activation: 2. Multiple aerosol
    types *J. Geophys. Res. Atmos.* **105**, 6837–6844. (2000).
Bush, M., Allen, T., Bain, C., Boutle, I., Edwards, J., Finnenkoetter, A., Franklin, C., Hanley, K.,
    Lean, H., Lock, A., et al. The first Met Office Unified Model–JULES
    Regional Atmosphere and Land configuration, RAL1 *Geosci. Model Dev.* **13**, 1999–2029.
    (2020).
Ghan, S. J., Abdul-Razzak, H., Nenes, A., Ming, Y., Liu, X., Ovchinnikov, M., Shipway, B.,
    Meskhidze, N., Xu, J. & Shi, X. Droplet nucleation: Physically-based parameterizations and
    comparative evaluation *J. Adv. Model. Earth Syst.* **3**, (2011).
Grosvenor, D. P., Field, P. R., Hill, A. A. & Shipway, B. J. The relative importance of
    macrophysical and cloud albedo changes for aerosol-induced radiative effects in closed-cell
    stratocumulus: insight from the modelling of a case study *Atmos. Chem. Phys.* **17**, 5155–
    5183. (2017).
Guichard, F. & Couvreux, F. A short review of numerical cloud-resolving models *Tellus, Ser. A
    Dyn. Meteorol. Oceanogr.* **69**, (2017).
Hill, A. A., Shipway, B. J. & Boutle, I. A. How sensitive are aerosol-precipitation interactions to
    the warm rain representation? *J. Adv. Model. Earth Syst.* **7**, 987–1004. (2015).
Khain, A. P., Beheng, K. D., Heymsfield, A., Korolev, A., Krichak, S. O., Levin, Z., Pinsky, M.,
    Phillips, V., Prabhakaran, T., Teller, A., et al. Representation of microphysical processes in
    cloud-resolving models: Spectral (bin) microphysics versus bulk parameterization *Rev.
    Geophys.* **53**, 247–322. (2015).
Lebo, Z. J. & Morrison, H. Dynamical Effects of Aerosol Perturbations on Simulated Idealized
    Squall Lines *Mon. Weather Rev.* **142**, 991–1009. (2014).
Lebo, Z. J., Morrison, H. & Seinfeld, J. H. Are simulated aerosol-induced effects on deep
    convective clouds strongly dependent on saturation adjustment? *Atmos. Chem. Phys.* **12**,
    9941–9964. (2012).
Malavelle, F. F., Haywood, J. M., Field, P. R., Hill, A. A., Abel, S. J., Lock, A. P., Shipway, B.

J. & McBeath, K. A method to represent subgrid-scale updraft velocity in kilometer-scale models: Implication for aerosol activation *J. Geophys. Res. Atmos.* **119**, 4149–4173. (2014).

Miltenberger, A. K., Field, P. R., Hill, A. A., Rosenberg, P., Shipway, B. J., Wilkinson, J. M., Scovell, R. & Blyth, A. M. Aerosol–cloud interactions in mixed-phase convective clouds – Part 1: Aerosol perturbations *Atmos. Chem. Phys.* **18**, 3119–3145. (2018).

Morrison, H. & Gettelman, A. A New Two-Moment Bulk Stratiform Cloud Microphysics Scheme in the Community Atmosphere Model, Version 3 (CAM3). Part I: Description and Numerical Tests *J. Clim.* **21**, 3642–3659. (2008).

Morrison, H. & Grabowski, W. W. Modeling Supersaturation and Subgrid-Scale Mixing with Two-Moment Bulk Warm Microphysics *J. Atmos. Sci.* **65**, 792–812. (2008).

Mulcahy, J., Johnson, C., Jones, C., Povey, A., Scott, C., Sellar, A., Turnock, S., Woodhouse, M., Andrews, M., Bellouin, N., et al. Description and evaluation of aerosol in UKESM1 and HadGEM3-GC3.1 CMIP6 historical simulations *Geosci. Model Dev. Discuss.* 1–59. (2020).

Phillips, V. T. J., Donner, L. J. & Garner, S. T. Nucleation Processes in Deep Convection Simulated by a Cloud-System-Resolving Model with Double-Moment Bulk Microphysics *J. Atmos. Sci.* **64**, 738–761. (2007).

Shipway, B. J. & Hill, A. A. Diagnosis of systematic differences between multiple parametrizations of warm rain microphysics using a kinematic framework *Q. J. R. Meteorol. Soc.* **138**, 2196–2211. (2012).

Smith, R. N. B. A scheme for predicting layer clouds and their water content in a general circulation model *Q. J. R. Meteorol. Soc.* **116**, 435–460. (1990).

Walko, R. L., Cotton, W. R., Feingold, G. & Stevens, B. Efficient computation of vapor and heat diffusion between hydrometeors in a numerical model *Atmos. Res.* **53**, 171–183. (2000).

West, R. E. L., Stier, P., Jones, A., Johnson, C. E., Mann, G. W., Bellouin, N., Partridge, D. G. & Kipling, Z. The importance of vertical velocity variability for estimates of the indirect aerosol effects *Atmos. Chem. Phys.* **14**, 6369–6393. (2014).

Williamson, C. J., Kupc, A., Axisa, D., Bilsback, K. R., Bui, T. P., Campuzano-Jost, P., Dollner, M., Froyd, K. D., Hodshire, A. L., Jimenez, J. L., et al. A large source of cloud condensation nuclei from new particle formation in the tropics *Nature*. **574**, 399–403. (2019).

Wilson, D. R. & Ballard, S. P. A microphysically based precipitation scheme for the UK meteorological office unified model *Q. J. R. Meteorol. Soc.* **125**, 1607–1636. (1999).

**Summary of main changes.**

Additional justification and clarifications concerning saturation adjustment

Perform and document, mainly in supplement, some tuned simulations to improve model-observation agreement as described in review replies.

Relabel activation schemes throughout for clarity

Merge section 6 and section 7 and exchange with section 5.

Substantial updates to model description and evaluation sections.

Additional table summarizing model configuration

Numerous clarifications in figure captions, including in the supplementary information.

Text changes are highlighted in red in the marked-up version attached.

[revised manuscript text omitted]
 liquid water content (LWC) with LWC $> 0.15\,\mathrm{g\,kg}^{-1}$ above the horizontal line and lower LWC below.

| Altitude (m) | $\bar{w}$ obs. ms$^{-1}$ | $\bar{w}$ model ms$^{-1}$ | $\sigma_w$ obs. ms$^{-1}$ | $\sigma_w$ 500 m ms$^{-1}$ |
|---|---|---|---|---|
| 800 | 0.69 | 0.98 | 1.32 | 0.90 |
| 1450 | 0.75 | 0.51 | 1.40 | 0.92 |
| 1900 | 0.72 | 0.17 | 1.47 | 0.73 |
| 2215 | 0.04 | 0.07 | 1.63 | 1.00 |
| 2550 | -0.61 | 0.17 | 2.80 | 1.37 |
| 800 | -0.006 | 0.12 | 0.88 | 0.34 |
| 1450 | -0.12 | -0.012 | 1.10 | 0.25 |
| 1900 | -0.88 | -0.04 | 0.92 | 0.22 |
| 2215 | -0.42 | -0.15 | 1.39 | 0.49 |
| 2550 | -0.90 | 0.52 | 2.35 | - |

[Figure]

**Figure S4.** Driving model evaluation against Ascension Island radiosondes on 19 August. The global model is shown with the dashed line and the 7 km regional model with the dotted line. To reduce computational expense, the domain we simulate at 500 m resolution does not include Ascension Island.

**Table S3.** Change to simulated number mean droplet radius when the exponential size distribution 'exp' is replaced offline by the size distribution of Morrison and Gettelman (2008), labelled MG08, based on the same momentsand when it is replaced by a gamma distribution with $\mu = 5$, labelled '$\mu = 5$'.

| Altitude | $\bar{r}$ (exp) ($\mu$m) | $\bar{r}$ (MG08) ($\mu$m) | $\bar{r}$ ($\mu = 5$) ($\mu$m) |
|---|---|---|---|
| 800 | 2.10 | 3.44 | 3.29 |
| 1450 | 2.40 | 3.89 | 3.76 |
| 1900 | 2.68 | 4.39 | 4.21 |
| 2215 | 3.54 | 5.60 | 5.56 |
| 2550 | 4.20 | 6.06 | 6.59 |

[Figure]

**Figure S5.** Simulated Aitken (AIT) and accumulation-mode (ACC) aerosol concentrations at ambient temperature and pressure, as in Figure 4, except including the global model (GLM) and extended to high altitudes. The comparison to the CLARIFY PCASP and CPC observations at low altitudes is retained.

.

**Table S4.** Values of the velocity scaling factor $\sqrt{f/R}$ for a horizontal resolution of 500 m, a factor $f = 4$ correction to the vertical velocity variance suggested by Malavelle et al. (2014), and two boundary layer heights representative of the CLARIFY case study.

| Boundary layer type | $Z_{ml}$ | $\sqrt{f/R}\,(Z = 1800\text{m})$ | $\sqrt{f/R}\,(Z = 2200\text{m})$ |
|---|---|---|---|
| II | $0.5Z$ | 4.27 | 3.85 |
| III | $1.3Z$ | 2.81 | 2.64 |
| IV | $0.5Z$ | 4.27 | 3.85 |
| V (sc) | $0.5Z$ | 4.27 | 3.85 |
| V (cbl) | $Z$ | 3.09 | 2.87 |
| VI | $Z$ | 3.09 | 2.87 |
| VII | $0.5Z$ | 4.27 | 3.85 |

[Figure]

**Figure S6.** Mean and standard deviation of in-cloud vertical velocity $w$ in observations and in simulations, at the usual flight altitudes and in bins of liquid water content with boundaries at 0.08, 0.2, and 0.5gkg$^{-1}$

[revised manuscript text omitted]

We fixed this bug in the ARG (DIAG) activation scheme (from UM version 11.7) by redefining the minimum updraft in the PDF that is sampled as

$$w_{min} = \max(0, \bar{w} - 2\sigma_w) \tag{S2}$$

instead of zero, and the maximum as

$$w_{max} = \max(4\sigma_w, \bar{w} + 2\sigma_w) \tag{S3}$$

instead of $4\sigma_w$, to preserve existing behaviour as far as possible but still ensure that the PDF is still sampled in strong updrafts.

[Figure]

**Figure S14.** Correlation of updraft with cloud droplet number calculated with the UKCA activation scheme at low altitudes (below 900 m) for the 7 km-resolution simulation. At or close to cloud base, the updrafts can still be significant compared to the minimum updraft width, and consequently the PDF is not always fully sampled. All times are used (not just midday on 19 August) to increase the statistics available.